# RECEVAL: Evaluating Reasoning Chains via Correctness and Informativeness

**Archiki Prasad      Swarnadeep Saha      Xiang Zhou      Mohit Bansal**

UNC Chapel Hill

{archiki, swarna, xzh, mbansal}@cs.unc.edu

## Abstract

Multi-step reasoning ability is fundamental to many natural language tasks, yet it is unclear what constitutes a *good* reasoning chain and how to evaluate them. Most existing methods focus solely on whether the reasoning chain leads to the correct conclusion, but this answer-oriented view may confound reasoning quality with other spurious shortcuts to predict the answer. To bridge this gap, we evaluate reasoning chains by viewing them as *informal proofs* that derive the final answer. Specifically, we propose RECEVAL (**Re**asoning **C**hain **Eval**uation), a framework that evaluates reasoning chains via two key properties: (1) *correctness*, i.e., each step makes a valid inference based on information contained within the step, preceding steps, and input context, and (2) *informativeness*, i.e., each step provides new information that is helpful towards deriving the generated answer. We evaluate these properties by developing metrics using natural language inference models and $\mathcal{V}$-Information. On multiple datasets, we show that RECEVAL effectively identifies various error types and yields notable improvements compared to prior methods. We analyze the impact of step boundaries, and previous steps on evaluating correctness and demonstrate that our informativeness metric captures the expected flow of information in high-quality reasoning chains. Finally, we show that scoring reasoning chains based on RECEVAL improves downstream task performance.[1]

## 1 Introduction

The ability to reason is fundamental to many natural language processing tasks (Lin et al., 2019; Duan et al., 2020). A reasoning chain composes multiple reasoning steps together to accomplish an end task such as solving complex reasoning problems involving commonsense (Talmor et al., 2019; Huang et al., 2019; Aggarwal et al., 2021) and arithmetic (Hendrycks et al., 2021; Cobbe et al., 2021).

---

[1]Code: https://github.com/archiki/ReCEval

---

> **Context:** The moon is a kind of moon. Earth is a kind of planet. Moons orbit planets. Gravity causes orbits.
> **Question:** What keeps the Moon orbiting Earth?
> **Model-generated Step-by-Step Rationales:**
> - Step 1: [Moon is a kind of moon] and [earth is a kind of planet], so [the moon and earth are planets].
> - Step 2: [Gravity causes orbits], so [gravity causes moon to orbit earth].
> **Answer:** Earth's gravity.

Figure 1: Model-generated step-by-step reasoning from Entailment Bank (Dalvi et al., 2021). Reasoning errors include: incorrect step inference (requires inferring *'moon orbits earth'*), and incorrect inference and uninformative (*'moon is a planet'* does not help answer the question). Reasoning Content Units (RCUs) are shown as '[.]'.

Recent advances in scaling language models have led to emergent reasoning capabilities, whereby a model is able to generate a reasoning chain in a few-shot manner (Wei et al., 2022; Chowdhery et al., 2022; Kojima et al., 2022). In most previous works, a model's reasoning capability is judged by its performance on the end task (Huang and Chang, 2022). However, this evaluation alone is not ideal for understanding the reasoning ability of models, as it implies a narrow view of correctness solely based on the answer, and may confound the model's reasoning capabilities with unfaithful or spurious reasoning shortcuts leading to the correct answer (Creswell and Shanahan, 2022; Lyu et al., 2023; Turpin et al., 2023). Thus, it is desirable to complement answer-oriented evaluation with an intrinsic evaluation of the quality of reasoning chains.

For a more comprehensive evaluation, prior works use human-written reasoning chains from Entailment Bank (Dalvi et al., 2021), StrategyQA (Geva et al., 2021), etc., to develop supervised metrics that evaluate model-generated reasoning chains with respect to human-written ones (Clinciu et al., 2021; Welleck et al., 2022). However, this evaluation strategy may be infeasible

due to the time-consuming and expensive nature of obtaining human-written chains (Welleck et al., 2021; Tian et al., 2021; Han et al., 2022). Moreover, the effectiveness of reference-based evaluations heavily relies on the selection and coverage of gold chains, which may not be unique (Dalvi et al., 2021). Golovneva et al. (2023) took the first step towards reference-free evaluation of reasoning chains by developing metrics based on generic reasoning errors like redundancy, hallucination, etc. In this work, we further explore this direction with the goal of formalizing desired properties of reasoning chains and introducing additional metrics to assess these properties effectively.

To evaluate reasoning chains in a reference-free manner, we first define the characteristics of *good* reasoning chains. In particular, we view reasoning chains as *informal proofs* that lead to the final answer (Welleck et al., 2022; Jiang et al., 2023). While reasoning chains operate over natural language and may not adhere to the strict nature of formal proofs (Welleck et al., 2021), they serve a similar role in providing rationales for the final answer. Conceptually, *each step in a reasoning chain should make a valid inference towards deriving the answer by leveraging prior information (i.e., previous steps or input context)*. In this work, we formalize this concept and propose a framework, RE-CEVAL (**Re**asoning **C**hain **Eval**uation) that defines good reasoning chains based on two properties: (1) *Correctness:* Each step generates a valid inference based on the information present (a) within the step (*intra-step*) and (b) past information present in the input context or derived in previous steps (*inter-step*); and (2) *Informativeness:* Each step provides new information that is helpful towards deriving the final answer (§3). Fig. 1 contains an example where these properties are violated.

RECEVAL introduces a collection of reference-free metrics that measure the correctness and informativeness of reasoning chains (§4). To measure correctness, we decompose reasoning chains into fine-grained components called **R**easoning **C**ontent **U**nits (RCUs), representing specific claims (as shown in Fig. 1). We measure informativeness by computing the information gain from including each step in the reasoning chain towards the final answer. We develop these metrics using a combination of Natural Language Inference models (Bowman et al., 2015; Williams et al., 2018) and information-theoretic measures that rely on $\mathcal{V}$-

information (Xu et al., 2020; Hewitt et al., 2021).

We evaluate RECEVAL against multiple reference-free metrics (§6). Our meta-evaluation procedure is based on correlation with automatically perturbed and human-annotated errors in English reasoning chains from Entailment Bank (Dalvi et al., 2021), GSM-8K (Cobbe et al., 2021), and DROP (Dua et al., 2019) respectively. On Entailment Bank, our metrics exhibit the highest correlation for 5 out of 6 error types, e.g., significantly boosting correlation from $0.62 \rightarrow 0.89$ for hallucinations. Additionally, on GSM-8K, and DROP, our metrics improve correlation from $0.28 \rightarrow 0.36$, and $0.19 \rightarrow 0.22$ for the overall quality measure respectively, excelling in identifying 5 out of 7 error types. Next, we conduct an extensive analysis of our metrics, showcasing how RCUs facilitate the evaluation of correctness and how high-quality human-written reasoning chains typically exhibit a positive trend in information gain (§6.2). Finally, we demonstrate that selecting high-scoring chains based on RECEVAL enhances downstream task performance (§6.3).

In summary, our contributions are:
1. Introducing RECEVAL, a framework that evaluates reasoning chains based on two desired attributes: correctness and informativeness.
2. Proposing reference-free metrics to measure correctness and informativeness using NLI models and $\mathcal{V}$-information. These metrics effectively identify various errors and surpass prior methods in meta-evaluation.
3. Conducting a comprehensive study of our metrics, demonstrating that RECEVAL can improve the downstream performance of reasoning tasks.

## 2 Reasoning Chains: Preliminaries

In this section, we formally define the concepts of reasoning chains, RCUs, and $\mathcal{V}$-information.

**Reasoning Chain.** Given a natural language reasoning task, let $\mathcal{X}$ denote the input context describing the problem. We define a reasoning chain $\mathcal{R} = \{s^{(1)}, \cdots, s^{(n)}\}$ as a multi-step rationale, consisting of $n$ steps, used to arrive at a predicted answer $\hat{a}$. Chains can be human-written or model-generated (as in CoT prompting (Wei et al., 2022)).

**Reasoning Content Unit (RCU).** We assume each step $s^{(i)}$ contains one or more claims, which we refer to as *Reasoning Content Units* (RCUs),

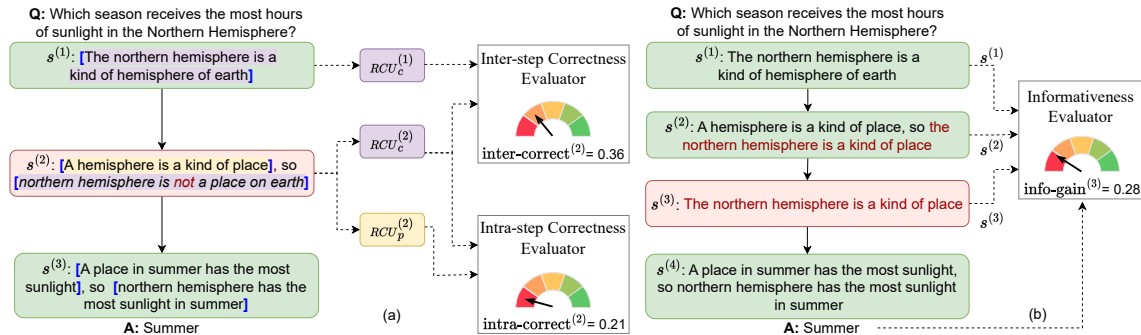

Figure 2: Evaluation of a reasoning chain using the RECEVAL framework: (a) Correctness of the second step using intra-correct$_{\text{entail}}$ and inter-correct metrics. Each step is divided into premise-RCUs and conclusion-RCU, denoted by '[.]'. (b) Informativeness of the third step in relation to preceding steps using info-gain$_{\text{PVI}}$ (see §4).

shown in Fig. 2 via '[.]'. RCUs are conceptually similar to Summary Content Units (SCUs) used in fine-grained summary evaluation (Nenkova and Passonneau, 2004; Shapira et al., 2019; Zhang and Bansal, 2021). Visualizing a reasoning chain as a sequence of steps and a step as a group of RCUs allows for fine-grained analysis and verification of a model's reasoning abilities. The RCUs in a step $s^{(i)}$ typically can be split into a single conclusion-RCU, denoted by $\text{RCU}_c^{(i)}$, and $t$ other premise-RCUs, denoted by $\text{RCU}_p^{(i)} = \{\text{RCU}_{p_j}^{(i)}\}_{j=1}^t$, where $t \geq 0$. For example, in Fig. 2(a), step $s^{(3)}$ contains two RCUs: the first ("a place ... most sunlight") is the premise, and the second ("northern ... in summer") is the conclusion. We discuss how to identify RCUs in §4.4 and their usefulness to RECEVAL in §6.2.

**Pointwise $\mathcal{V}$-Information (PVI).** In this paper, we utilize $\mathcal{V}$-Information, an information-theoretic concept that we introduce briefly here (with additional details in Appendix A). Given two random variables $X$ and $Y$, Xu et al. (2020) propose an empirical approximation of the conditional entropy $H_\mathcal{V}(Y|X)$ via a family of models $\mathcal{V}$ that estimates their probability distribution. Thus, we compute the amount of information in $X$ about $Y$ as:

$$I_\mathcal{V}(X \rightarrow Y) = H_\mathcal{V}(Y|\varnothing) - H_\mathcal{V}(Y|X)$$

Ethayarajh et al. (2022) propose *pointwise $\mathcal{V}$-information (PVI)* to measure the degree of usable information present in individual data points $(x, y)$:

$$\text{PVI}(x \rightarrow y) = -\log g'[\varnothing](y) + \log g[x](y)$$

using trained models $g, g' \in \mathcal{V}$. These models take $x$ or $\varnothing$ (e.g., empty string) as input to yield the probability of generating $y$. This extends to conditional PVI relative to an instance $z$ as:

$$\text{PVI}(x \rightarrow y|z) = -\log g'[z](y) + \log g[z, x](y)$$

Unless mentioned otherwise, we use T5-large (Raffel et al., 2020) as our model family $\mathcal{V}$.

## 3 Properties of Good Reasoning Chains

Reasoning chains are informal proofs leading to the final answer. We propose evaluating their quality based on *correctness* and *informativeness*.

**Correctness.** For a reasoning chain to be correct, every step must be correct. Further, we say a step $s^{(i)}$ is correct if its corresponding conclusion $\text{RCU}_c^{(i)}$ is correct. Two factors contribute to step correctness: (1) *intra-step correctness*, which evaluates if $\text{RCU}_c^{(i)}$ is correct based on the premise units $\text{RCU}_p^{(i)}$ within the step, and (2) *inter-step correctness*, which evaluates if $\text{RCU}_c^{(i)}$ is correct given the previous context (input $\mathcal{X}$ and previous steps $s^{(<i)}$). Intuitively, intra-step correctness evaluates consistency of claims within the step, while inter-step correctness measures global consistency. In Fig. 2(a), $\text{RCU}_c^{(2)}$ in $s^{(2)}$ does not follow from $\text{RCU}_p^{(2)}$, incorrectly concluding that northern hemisphere is not a place on earth and also contradicts $\text{RCU}_c^{(1)}$.

**Informativeness.** In addition to correctness, we also evaluate the complementary property of *informativeness*. This property measures the helpfulness and importance of each reasoning step in producing the final answer. Not all (plausible) inferences made in a step are equally relevant to the question at hand, so informativeness captures how much a particular step contributes towards getting closer to the answer. Fig. 2(b) demonstrates the role of informativeness. While the third step $s^{(3)}$ does not alter correctness, it also does not move us closer to the answer beyond the second step. Thus, evaluating reasoning based on informativeness helps identify issues such as repetition or redundancy.

Next, we describe the technical details of our

metrics that evaluate every reasoning step by itself (intra-step correctness), how it relates to the input and prior steps (inter-step correctness), and how it aids in solving the problem (informativeness).

# 4 RECEVAL: Evaluation Metrics

We now introduce our RECEVAL (**Re**asoning **C**hain **Eval**uation) framework that builds upon the desired properties of reasoning chains.

## 4.1 Evaluation of Intra-Step Correctness

We propose two methods to measure the intra-step correctness of a reasoning step based on two complementary views of correctness.

**Entailment-based Intra-Step Correctness.** Our first method aims to capture correctness by computing the entailment probability of the conclusion-RCU ($\text{RCU}_c^{(i)}$) given the premise-RCUs ($\text{RCU}_p^{(i)}$) within a step $s^{(i)}$ as follows:

$$\text{intra-correct}_{\text{entail}}^{(i)} = P_{\text{entail}}(\text{RCU}_p^{(i)}; \text{RCU}_c^{(i)})$$

The premise-RCUs are concatenated and the entailment probability $P_{\text{entail}}$ is computed using an off-the-shelf NLI model (Laurer et al., 2022). We *strictly* define entailment, whereby a conclusion-RCU neutral to the premise-RCUs receives a low probability. This design choice accounts for incorrect reasoning steps that may contain hallucinations or unsupported non-factual claims.

**PVI-based Intra-Step Correctness.** Our previous method requires strict entailment between premise-RCUs and conclusion-RCU. However, in natural language, reasoning steps can be informal and still be considered correct with some premise-RCUs omitted. To allow for such flexibility, we introduce a relaxed criterion that evaluates the ease of drawing a conclusion from the premise. Using PVI (introduced in §2), we evaluate the ease of generating a conclusion-RCU based on the useful information already present in the premise-RCUs. Formally, we express our metric as:

$$\text{intra-correct}_{\text{PVI}}^{(i)} = \text{PVI}(\text{RCU}_p^{(i)} \rightarrow \text{RCU}_c^{(i)})$$

## 4.2 Evaluation of Inter-Step Correctness

The aforementioned methods assess local correctness based on premise-RCUs within a step. In reasoning chains with numerous steps, it is crucial to ensure that any new conclusion-RCU remains consistent with *all* known information, whether in the input $\mathcal{X}$ or in all prior conclusion-RCUs $\text{RCU}_c^{(<i)}$. To measure this 'global' inter-step correctness, we

verify the absence of contradictions between the current $\text{RCU}_c^{(i)}$ and prior information, including $\mathcal{X}$ and $\text{RCU}_c^{(<i)}$. For example, Fig. 2(a) for step $s^{(2)}$, we evaluate the consistency of $\text{RCU}_c^{(2)}$ with $\text{RCU}_c^{(1)}$. Similar to §4.1, we utilize an NLI model to obtain the contradiction probability ($P_{\text{contr.}}$), to calculate:

$$\text{inter-correct}^{(i)} = 1 - max_r(P_{\text{contr.}}(r; \text{RCU}_c^{(i)}))$$

where, $r \in \mathcal{X} \cup \{\text{RCU}_c^{(j)}\}_{j=1}^{i-1}$. We evaluate only conclusion-RCUs, excluding premise-RCUs from prior steps due to their overlap with input context $\mathcal{X}$. Empirically, we verify that excluding premise-RCUs does not impact performance.

## 4.3 Evaluation of Informativeness

As mentioned in §2, a good reasoning chain not only ensures correctness but also promotes informativeness towards the final answer. To compute this metric, we employ conditional PVI (see §2).

**PVI-based Information Gain.** In order to capture the contribution of a reasoning step, we measure the gain in information after adding it to the chain (constructed so far). A large positive gain indicates that the step makes predicting the answer easier. For instance, the low value of information gain of step $s^{(3)}$ in Fig. 2(b) suggests that the step is redundant. Inspired by Chen et al. (2022), who use conditional PVI relative to the question and gold answer, we compute information provided by a step $s^{(i)}$ toward the predicted answer $\hat{a}$, conditioned on the previous steps $s^{(<i)}$, denoted as:

$$\text{info-gain}_{\text{PVI}}^{(i)} = \text{PVI}(s^{(i)} \rightarrow \hat{a}|s^{(<i)})$$

## 4.4 RECEVAL: Overall Algorithm

We now describe our overall RECEVAL algorithm based on the aforementioned step-level metrics.

**Identifying RCUs.** We begin by splitting each step into constituent RCUs using an off-the-shelf Semantic Role Labeling (SRL) model that decomposes sentences into semantic triplets with 'subject-verb-object' frames (Shi and Lin, 2019; Zhang and Bansal, 2021). Multiple frames are generated for each sentence, from which we extract non-overlapping frames as our units. These extracted RCUs within each step are classified as premise or conclusion RCUs based on their location within the sentence and sentence structure (see Appendix A).

**Overall Reasoning Chain Evaluation.** After decomposing a step into RCUs, we assess their correctness and informativeness using the metrics outlined in §4. The step-level evaluations are

then combined to determine the overall quality of the reasoning chain. Following Golovneva et al. (2023), we posit a reasoning chain is only as good as its least correct or least informative step, i.e., for each metric we use 'min' aggregation across steps (see Algorithm 1 in Appendix A). These chain-level scores for each metric facilitate the identification of different error types (results in §6).

Additional implementation details of RECEVAL including model checkpoints, identifying RCUs, and computing PVI are present in Appendix A.

## 5 Meta-Evaluation Setup

We evaluate a metric's ability to detect errors in reasoning chains using the meta-evaluation framework used by Golovneva et al. (2023). For each error category, we compute the correlation between ground-truth annotations (§5.1) and metrics (§5.2).

### 5.1 Meta-Evaluation: Datasets

We use three datasets, Entailment Bank (EB), GSM-8K, and DROP to evaluate RECEVAL. EB is a deductive reasoning dataset containing multi-step reasoning chains. Golovneva et al. (2023) emulate reasoning errors on EB via programmatic perturbations (henceforth referred to as EB-regular) creating errors such as hallucinations (HALL), negation (NEG), swap (SWAP), verbatim repetition (REP). Conversely, using the same error categories, we generate more realistic and challenging errors by applying perturbations on intermediate inferences (referred to as EB-challenge). This also includes interesting variations of informativeness errors such as adding a paraphrase of a step (PAR), or a sentence irrelevant to the reasoning problem (RED). In both versions, we consider only one error at a time.

GSM-8K contains grade school math word problems requiring mathematical reasoning. We evaluate model-generated CoT steps (Wei et al., 2022) using human judgments from Golovneva et al. (2023). DROP (Dua et al., 2019) contains discrete reasoning questions over a paragraph. We evaluate reasoning chains generated by Golovneva et al. (2023) using GPT-3 (Brown et al., 2020) against human judgement annotations. These annotations include evaluations for factuality issues (FACT), logical deduction errors (LOGIC), hallucinations (HALL), redundant or irrelevant information (RED), unnecessary paraphrasing (REP), commonsense errors (COM), and arithmetic errors (MATH). Furthermore, the dataset contains two overall scores mea-

suring the quality (QUAL) and coherence (COH) of the reasoning chain on a Likert scale. Note that in GSM-8K and DROP, a single model-generated reasoning chain can contain multiple errors.

For a summary of errors, refer to Table 19 (Appendix B). Additional details about both datasets including examples are also present in Appendix B.

### 5.2 Meta-Evaluation: Baselines

Following Golovneva et al. (2023), we choose baseline text-generation metrics measuring $n$-gram match (ROUGE-2 Lin (2004)), and model-based metrics such as BERTScore (Zhang* et al., 2020), BARTScore (Yuan et al., 2021), and CTC (Deng et al., 2021). Each metric compares the reasoning chain $\mathcal{R}$ (as a paragraph) with the input context $\mathcal{X}$. We also compare against semantic similarity (SS), alignment (SA), and logical inference (LI) metrics from ROSCOE. For ROSCOE-SA, and -SS, we use the fine-tuned text-similarity models (Golovneva et al., 2023). We further group the reference-free metrics from ROSCOE that measure redundancy (repetition-token and -step) as ROSCOE-REP. This enables a direct comparison with ROSCOE on two desired properties: correctness and informativeness. To evaluate correctness, we compare with ROSCOE-SA, -SS, and -LI, while for informativeness, we compare with ROSCOE-SA, -SS, and -REP.

### 5.3 Meta-Evaluation: Correlation Measure

After scoring reasoning chains with either RECEVAL or baseline metrics, we evaluate whether the scores indicate the presence or absence of each error type. We again follow past work to employ Somer's-$D$ correlation (Somers, 1962), i.e., we assess a metric $S$ against the random variable denoting the chain's error status ($E \in 0, 1$). Somer's-$D$ correlation, computed using Kendall's $\tau$ coefficient, is defined as: $D_{SE} = \tau(E, S)/\tau(E, E)$. When multiple metrics are available (as in ROSCOE or RECEVAL), we compute the correlation with each variant and report the highest correlation obtained.

## 6 Results and Discussion

### 6.1 Effectiveness of RECEVAL

In this section, we present our main meta-evaluation results on EB, GSM-8K, and DROP.

**Entailment Bank.** Table 1 presents the meta-evaluation results for different error types in the EB-challenge dataset. Our RECEVAL metrics outperform text-generation baselines on both correct-

| Metric | HALL | NEG | SWAP |
|---|---|---|---|
| ROUGE-2 | -0.01 | -0.02 | 0.14 |
| BERTScore | 0.09 | 0.02 | 0.07 |
| BARTScore | 0.00 | -0.01 | 0.07 |
| CTC | 0.09 | -0.04 | -0.05 |
| ROSCOE-SA | 0.62 | 0.40 | 0.22 |
| ROSCOE-SS | 0.34 | 0.40 | 0.09 |
| ROSCOE-LI | 0.20 | 0.82 | 0.16 |
| RECEVAL-correctness | **0.89** | **0.88** | **0.39** |

(a) Correctness

| Metric | REP | PAR | RED |
|---|---|---|---|
| ROUGE-2 | 0.43 | 0.21 | 0.11 |
| BERTScore | 0.24 | 0.16 | 0.12 |
| BARTScore | 0.11 | 0.12 | 0.08 |
| CTC | 0.24 | 0.14 | 0.10 |
| ROSCOE-SA | **0.83** | 0.64 | 0.51 |
| ROSCOE-SS | 0.81 | 0.62 | 0.54 |
| ROSCOE-REP | **0.83** | 0.64 | 0.48 |
| RECEVAL-informativeness | 0.66 | **0.68** | **0.67** |

(b) Informativeness

Table 1: Meta-evaluation (Somer's D) on EB-challenge (test). Table 16 in Appendix C shows similar trends on EB-regular. We **bold** the highest and underline the second-highest correlation values (higher correlation is better).

| Metric | QUAL | COH | COM | FACT | HALL | RED | REP | LOGIC | MATH |
|---|---|---|---|---|---|---|---|---|---|
| ROUGE-2 | 0.09 | 0.14 | 0.06 | 0.10 | 0.17 | -0.02 | 0.56 | 0.03 | 0.11 |
| BERTScore | 0.19 | 0.23 | 0.12 | 0.13 | 0.20 | 0.13 | 0.94 | 0.15 | 0.13 |
| BARTScore | 0.01 | 0.03 | -0.05 | 0.04 | -0.25 | -0.26 | 0.42 | 0.00 | -0.55 |
| CTC | -0.09 | -0.15 | -0.08 | -0.11 | 0.01 | -0.37 | 0.57 | -0.11 | -0.09 |
| ROSCOE-SA | 0.20 | 0.19 | 0.19 | 0.08 | 0.22 | 0.39 | 0.79 | 0.18 | **0.44** |
| ROSCOE-SS | 0.20 | 0.17 | 0.17 | 0.14 | 0.25 | 0.51 | 0.87 | 0.15 | 0.23 |
| ROSCOE-LI | 0.28 | 0.26 | 0.18 | 0.34 | 0.22 | 0.35 | **0.98** | 0.22 | 0.09 |
| ROSCOE-REP | 0.20 | 0.19 | 0.19 | 0.14 | 0.25 | 0.51 | 0.87 | 0.18 | **0.44** |
| RECEVAL-correctness | **0.36** | **0.31** | **0.21** | **0.37** | **0.28** | 0.40 | 0.63 | **0.25** | 0.24 |
| RECEVAL-informativeness | 0.30 | 0.29 | 0.19 | 0.26 | 0.26 | **0.55** | 0.87 | 0.21 | 0.32 |

Table 2: Meta-evaluation (Somer's D) on GSM-8K (test) with human-annotated errors from Golovneva et al. (2023).

ness and informativeness-based errors by up to $0.09 \rightarrow 0.89$, and $0.21 \rightarrow 0.68$ respectively. In terms of correctness, Table 1a shows that RECEVAL outperforms ROSCOE improving correlation from $0.62 \rightarrow 0.89$, and $0.22 \rightarrow 0.39$ on hallucinations, and swap errors respectively. For informativeness, from Table 1b, we observe that RECEVAL outperforms all baselines for complex errors like paraphrasing and redundancy by at least $0.64 \rightarrow 0.68$ and $0.54 \rightarrow 0.67$ respectively. While RECEVAL yields higher correlation compared to text-generation metrics for verbatim repetition (REP), ROSCOE achieves the best performance. Similar trends are observed in the evaluation on EB-regular, as shown in Table 16 in Appendix C.

**GSM-8K.** Table 2 shows the meta-evaluation results for GSM-8K. RECEVAL outperforms baseline metrics on the majority of error types. Compared to text-generation metrics, we achieve higher correlations across all error types. Notably, our metrics show higher correlations on overall quality (QUAL) and coherence (COH), outperforming ROSCOE-LI and ROSCOE semantic metrics by up to $0.28 \rightarrow 0.36$ and $0.20 \rightarrow 0.36$ respectively. We also obtain higher correlations on commonsense

(COM), factuality (FACT), hallucination (HALL), and logical (LOGIC) errors by up to $0.06$. In terms of informativeness, our metric yields highest correlation on RED and performs comparably to ROSCOE on REP errors. Our metrics are not specifically designed for arithmetic errors, which can be better handled using calculators or ROSCOE-REP. However, we leave this study for future work.

**DROP.** We observe similar trends on the DROP dataset, shown in Table 3, even though it primarily consists of single-step rationales ($< 20\%$ rationales are multi-step). RECEVAL outperforms all the baseline text-generation metrics and achieves matching if not better correlations compared to ROSCOE on overall QUAL and COH measures. Specifically, we obtain higher correlations on commonsense, factuality, hallucination, and logical errors by up to $0.08$. Additionally, we also improve correlations on RED errors when compared to ROSCOE ($0.80 \rightarrow 0.83$).

## 6.2 Analysis of RECEVAL Metrics

We analyze our RECEVAL metrics on EB dataset by addressing the following research questions.

| Metric | QUAL | COH | COM | FACT | HALL | RED | REP | LOGIC |
|---|---|---|---|---|---|---|---|---|
| ROUGE-2 | 0.14 | -0.15 | 0.49 | 0.32 | -0.28 | -0.72 | -0.44 | 0.03 |
| BERTScore | 0.13 | -0.12 | 0.49 | 0.28 | -0.18 | -0.65 | -0.04 | 0.00 |
| BARTScore | -0.09 | -0.39 | 0.58 | 0.16 | -0.45 | -0.84 | -0.89 | -0.23 |
| CTC | -0.03 | -0.10 | -0.07 | 0.33 | -0.04 | -0.62 | -0.09 | -0.09 |
| ROSCOE-SA | 0.19 | -0.31 | 0.44 | 0.51 | -0.06 | -0.57 | -0.60 | 0.10 |
| ROSCOE-SS | 0.11 | **0.36** | 0.46 | 0.22 | 0.16 | 0.80 | **0.91** | 0.05 |
| ROSCOE-LI | 0.20 | 0.24 | 0.46 | 0.39 | -0.01 | 0.08 | 0.70 | 0.01 |
| ROSCOE-REP | 0.07 | **0.36** | -0.14 | 0.17 | 0.45 | 0.80 | **0.91** | 0.05 |
| RECEVAL-correctness | **0.22** | 0.32 | **0.52** | **0.54** | **0.49** | 0.21 | -0.12 | **0.16** |
| RECEVAL-informativeness | 0.20 | **0.36** | 0.14 | 0.51 | 0.48 | **0.83** | 0.89 | 0.12 |

Table 3: Meta-evaluation (Somer's D) on DROP (test) with human-annotated errors from Golovneva et al. (2023).

| Method | intra-correct | | | inter-correct | | |
|---|---|---|---|---|---|---|
| | HALL | NEG | SWAP | HALL | NEG | SWAP |
| w/o RCUs | - | - | - | 0.12 | 0.83 | 0.11 |
| our RCUs | 0.71 | 0.84 | 0.37 | 0.14 | 0.90 | 0.16 |
| gold RCUs | 0.89 | 0.94 | 0.54 | 0.16 | 0.96 | 0.16 |

Table 4: Comparison of correctness metrics in RECEVAL on EB-challenge (dev split) with different RCU selection. Specifically, we use intra-correct$_{entail}$.

| Metric | HALL | NEG | SWAP |
|---|---|---|---|
| inter-correct ($k = 1$) | 0.08 | 0.79 | 0.14 |
| inter-correct ($k = 2$) | 0.10 | 0.84 | 0.17 |
| inter-correct ($k = all$) | 0.14 | 0.90 | 0.16 |

Table 5: Comparison of inter-correct metric with varying prior information (number of preceding steps denoted by $k$) on dev split of EB-challenge.

| Step Granularity | HALL | NEG | SWAP |
|---|---|---|---|
| Step = RCU | 0.46 | 0.87 | 0.28 |
| Step = sentence (as in RECEVAL) | 0.86 | 0.90 | 0.38 |
| Step = $\mathcal{R}$ | 0.17 | 0.32 | 0.13 |

Table 6: Comparing correctness metrics in RECEVAL for varying step boundaries on EB-challenge (dev split).

**How do RCU design choices affect correctness evaluation?** We examine the impact of different RCU design choices on correctness metrics (§4). We compare variants using (i) identified RCUs, (ii) no RCUs (treating a step as a whole), and (iii) gold RCU annotations (oracle setting). Gold RCUs are extracted using reasoning trees from the EB dataset (details in Appendix D). Results in Table 4 show the crucial role of RCU decomposition in RECEVAL, enabling accurate identification of hallucinations and swap errors. Gold RCUs improve correctness metrics and yield higher correlation across errors (up to 0.20). Nevertheless, our identified RCUs bolster correctness evaluation, and future work can bridge the gap between the two settings.

**How does the amount of previous information impact inter-step correctness?** In inter-step correctness (§4.2), we evaluate if a given step contradicts any conclusion-RCUs from prior steps or the input context $\mathcal{X}$. We explore the impact of prior information on inter-step correctness by considering $k$ preceding steps. We analyze three variants with $k = 1, 2$, and $all$ in Table 5. We observe that using only immediately preceding steps (i.e., $k = 1, 2$) leads to a decrease in correlation by up to 0.11 for hallucination and negate errors. Thus, evaluating inter-step correctness with respect to all previous steps is crucial for identifying potential errors.

**What constitutes a step and how does its granularity impact RECEVAL's effectiveness?** Un-like formal proofs, it is not straightforward to demarcate the step boundaries in natural language reasoning chains. To demonstrate the impact of step boundaries on reasoning evaluation, in Table 6 we compare three settings: (i) each RCU as a step, (ii) each sentence as a step, and (iii) the entire reasoning chain as a single step. Both extreme boundaries lead to decreased correlation across errors. RCU-level boundaries result in lower correlations on HALL and SWAP errors. Treating the entire chain as a step yields lower correlations on all errors, focusing only on the final conclusion. Hence, choosing appropriate step boundaries is crucial for evaluating multi-step rationales, and considering each sentence as a step proves effective in practice.

**How does informativeness vary across steps?** To further test our informativeness metric, we investigate whether human-written reasoning chains exhibit positive information gain for each step, and how they compare to chains with uninformative steps. We note that even for good reasoning chains, each step individually may not always be more informative than the previous step but approximately, a collection of every few consecutive steps

| Chain | $k=1$ | $k=2$ | $k=3$ |
|---|---|---|---|
| Uninformative (REP) | 36.4 | 69.4 | 80.7 |
| Uninformative (PAR) | 35.3 | 70.5 | 81.4 |
| Uninformative (RED) | 38.6 | 73.4 | 82.8 |
| Gold | 72.7 | 87.7 | 92.0 |

Table 7: % of $\text{API}_k$ chains in dev split of EB-challenge.

| Model | Method | REP | PAR | RED |
|---|---|---|---|---|
| GPT-2 XL (1.5B) | info-gain$_{\text{PVI}}$ | 0.67 | 0.66 | 0.65 |
| | info-gain$_{\text{LL}}$ | 0.58 | 0.60 | 0.60 |
| LLaMA-7B | info-gain$_{\text{LL}}$ | 0.69 | 0.70 | 0.68 |

Table 8: Comparison of info-gain metric using trained PVI models and pretrained LMs on EB-challenge (dev).

should show such behavior. Thus, we introduce a metric called *Approximately Positive Information-gain* (API). We say that for a reasoning chain $\mathcal{R}$, $\text{API}_k(\mathcal{R})=1$, if for every $k$ consecutive steps in the chain, these steps as a single unit are more informative than the preceding step. Formally, this is defined as $\sum_{j=i}^{i+k-1}\text{info-gain}_{\text{PVI}}^{(j)}>0, \forall s^{(i)} \in \mathcal{R}$ and 0 otherwise. Table 7 shows that 72% of gold chains have positive information-gain for all steps (i.e., $\text{API}_1=1$), considerably higher than uninformative chains (38%). We also observe that 87% of gold reasoning chains have positive gains for two consecutive steps (i.e., $\text{API}_2=1$), and as high as 92% for three consecutive steps (i.e., $\text{API}_3=1$). Thus, almost all high-quality reasoning chains demonstrate (approximately) positive information gain which is effectively captured by our info-gain$_{\text{PVI}}$ metric. It is also able to distinguish between informative and uninformative chains. Further analysis of informativeness trends is present in Appendix E.

**How does the underlying probability model affect info-gain?** In §4.3, computing conditional PVI requires fine-tuned models to learn text distributions from reasoning steps. In the absence of gold reasoning steps for training, we propose an alternative called info-gain$_{\text{LL}}$ that computes log-likelihood of steps directly from a pretrained LM like GPT-2 XL.[2] Comparing both approaches in Table 8, we find that info-gain$_{\text{PVI}}$ achieves higher correlations (by at least 0.05) across errors. Although fine-tuned LMs are more effective, the corresponding pretrained LMs can also be used to measure informativeness. However, using a larger pretrained LM such as LLaMA-7B (Touvron et al., 2023) can

---

[2]We use GPT-2 XL instead of T5-large as the latter is not an auto-regressive LM and cannot reliably be used to estimate log-likelihood without finetuning.

| Method | Error Types | | |
|---|---|---|---|
| inter-correct | HALL | NEG | SWAP |
| w/ NLI Model | 0.89 | 0.88 | 0.39 |
| w/ GPT-3.5-turbo | 0.86 | 0.91 | 0.38 |
| info-gain$_{\text{LL}}$ | REP | PAR | RED |
| w/ GPT-2 XL | 0.50 | 0.56 | 0.53 |
| w/ GPT-3.5-turbo | 0.54 | 0.58 | 0.56 |

Table 9: Using prompted LLM GPT-3.5 turbo to compute inter-step correctness (top) and informativeness (bottom) metrics on 50 dev instances from EB.

more than compensate for this performance gap, achieving the highest correlation in Table 8.

### 6.3 Utilizing RECEVAL for Evaluating and Improving Downstream Tasks

**Applying RECEVAL in Diverse Scenarios.** We consolidate our findings with different models and sub-metrics by making some recommendations on how to use RECEVAL in various evaluation settings. We sugggest using the NLI model by Laurer et al. (2022) for evaluating correctness, as it consistently performs well. For evaluating informativeness in tasks with gold reasoning chains, like EB, we advise using a T5-Large model. This choice aligns with other automatic metrics in (Chen et al., 2022; Golovneva et al., 2023). Otherwise, when gold reasoning chains are unavailable, we suggest opting for a larger pretrained LM like LLaMA-7B.

**Recent results on using GPT-3.5 with RECEVAL.** Some recent works focus on using large language models (LLMs) for evaluating text-generation outputs (Fu et al., 2023a; Liu et al., 2023) and self-verification (Kadavath et al., 2022; Ling et al., 2023). Inspired by this, we conduct a small-scale study to investigate if prompted LLMs, such as GPT-3.5-turbo (Ouyang et al., 2022), can be incorporated within RECEVAL on a subset of 50 reasoning chains from the EB dataset. To measure correctness and informativeness, we prompt the model to output a real-valued score between 0 to 1 as the probability of entailment and the probability of generating the answer respectively (details in Appendix A). Table 9 shows that instead of using pretrained models for which logits are available, we can also extend RECEVAL by prompting state-of-the-art LLMs such as GPT-3.5-turbo. We underscore that the core concept of evaluating for correctness and informativeness remains robust and general, regardless of the underlying LM used –

| Method | Accuracy (%) |
|---|---|
| Greedy Decoding | 17.3 |
| Sampling + ROSCOE (LI) | 19.0 |
| Sampling + ROSCOE (SA, SS) | 17.8 |
| Sampling + ROSCOE (REP) | 18.6 |
| Sampling + RECEVAL (correctness) | 19.6 |
| Sampling + RECEVAL (informativeness) | 18.7 |
| Sampling + RECEVAL (both) | 20.5 |

Table 10: Applying RECEVAL to improve downstream task performance on GSM-8K using FLAN T5-XXL.

even as more advanced models emerge.

**RECEVAL improves Downstream Task Performance.** Finally, we also examine if higher-quality reasoning chains (ranked using our metrics) yield improvements in downstream task performance with CoT prompting. To this end, generate reasoning chains for GSM-8K using FLAN T5-XXL (Chung et al., 2022). We sample 20 reasoning chains that are scored using metrics from RECEVAL or ROSCOE, and we select the chain with the lowest cumulative rank (details in Appendix A). We compare with ROSCOE in three settings: (i) ROSCOE-LI (best performance on overall measures in Table 2), (ii) ROSCOE-REP (analogous to informativeness), and (iii) non-repetition metrics from ROSCOE-SA and ROSCOE-SS (analogous to correctness).[3] Table 10 shows that RECEVAL improves QA accuracy by 3.2% over greedy decoding when considering both correctness and informativeness. Using only correctness or informativeness leads to improvements of 2.3% and 1.4%, respectively. In comparison, different combinations of ROSCOE metrics improve accuracy by up to 1.7%. This highlights a complementary benefit of evaluation metrics for reasoning chains. Further research can explore combining these metrics with other sampling strategies (Wang et al., 2023; Fu et al., 2023b) to enhance the reasoning capability of LLMs.

## 7    Related Work

Traditional text generation evaluation metrics use *n*-gram overlap (Papineni et al., 2002; Lin, 2004; Banerjee and Lavie, 2005), embeddings (Zhao et al., 2019; Zhang* et al., 2020; Sellam et al., 2020), information alignment (Deng et al., 2021), paraphrases (Thompson and Post, 2020), or text-generation models (Yuan et al., 2021; Fu et al.,

2023a), and are suitable for comparing machine-generated text to target text in tasks like summarization and machine translation. However, they are inadequate for evaluating reasoning chains with a coherent sequence of steps leading to the final answer. Additionally, relying on references makes them unsuitable for reference-free evaluation.

Some prior works on evaluating reasoning chains propose metrics based on specific construction and domain of datasets, making them less generalizable. For example, FOLIO (Han et al., 2022) and PrOntoQA (Saparov and He, 2023) use a fixed grammar to convert natural language reasoning chains to symbolic proofs that are evaluated using gold proofs. Dalvi et al. (2021) compare model-generated reasoning trees to gold reasoning trees. Closest to our work, Golovneva et al. (2023) proposed ROSCOE, a suite of reference-free and reference-based metrics that measure semantic alignment, similarity, and logical inference in reasoning chains. Building upon their work, we first formally define desired properties of good reasoning chains (i.e., correctness and informativeness) and then propose reference-free metrics (using RCUs and $\mathcal{V}$-information) that outperform ROSCOE across datasets.

## 8    Conclusion

We present RECEVAL, a framework for evaluating reasoning chains based on correctness and informativeness. We propose reference-free metrics for measuring these properties that are based on entailment and PVI, leveraging granular claims in reasoning chains called Reasoning Content Units (RCUs). Our approach considerably outperforms previous baseline metrics, as shown by meta-evaluation on multiple datasets. We also perform detailed analysis of our metrics and demonstrate that RECEVAL is effective in various settings, and leads to downstream improvement in task performance.

## Acknowledgements

We thank the reviewers and the area chairs for their helpful comments. We also thank Peter Hase, Prateek Yadav, and Shiyue Zhang for their feedback. This work was supported by NSF-CAREER Award 1846185, NSF-AI Engage Institute DRL-2112635, DARPA Machine Commonsense (MCS) Grant N66001-19-2-4031, and a Google Ph.D. Fellowship. The views contained in this article are those of the authors and not of the funding agency.

---

[3]We did not observe further accuracy improvements by combining all ROSCOE metrics.

## Limitations

An interesting assumption for future work to address is that all knowledge typically needed to evaluate the correctness of a reasoning step is explicitly present as part of the input or the intermediate reasoning steps. In scenarios where correctness depends on implicit knowledge, we rely on the choice of underlying models (described in Appendix A) which are built on top of pre-trained LMs and are known to capture a lot of background knowledge (Petroni et al., 2019; Roberts et al., 2020). However, inferences that rely on substantial implicit knowledge may not be best evaluated through current metrics. While current evaluation frameworks focus on evaluating the quality of model-generated reasoning chains, Wei et al. (2022) note that the chain itself may not faithfully reflect the internal reasoning process of the model. This remains an open question for future work to address.

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

---

**Algorithm 1** Chain-level Scores in RECEVAL

1: Input: Context $\mathcal{X}$, Reasoning Chain $\mathcal{R}$, Predicted Answer $\hat{a}$
2: Output: Overall scores for $\mathcal{R}$ with each metric
3: **for** $s^{(i)} \in \mathcal{R}$ **do**
4: $\quad \text{RCU}_{\boldsymbol{p}}^{(i)}, \text{RCU}_c^{(i)} \leftarrow \text{content-units}(s^{(i)})$
5: $\quad \text{score}_{\text{intra}}^{(i)} \leftarrow \text{intra-correct}^{(i)}(\text{RCU}_{\boldsymbol{p}}^{(i)}, \text{RCU}_c^{(i)})$
6: $\quad \text{score}_{\text{inter}}^{(i)} \leftarrow \text{inter-correct}^{(i)}(\text{RCU}_c^{(i)}, \mathcal{X}, s^{(<i)})$
7: $\quad \text{score}_{\text{info}}^{(i)} \leftarrow \text{info-gain}_{\text{PVI}}^{(i)}(s^{(\leq i)}, \hat{a})$
8: **end for**
9: $\text{score}_{\text{intra}} = \min_{i \in [1,n]}(\text{score}_{\text{intra}}^{(i)})$
10: $\text{score}_{\text{inter}} = \min_{i \in [1,n]}(\text{score}_{\text{inter}}^{(i)})$
11: $\text{score}_{\text{info}} = \min_{i \in [1,n]}(\text{score}_{\text{info}}^{(i)})$
12: **return** $\text{score}_{\text{intra}}, \text{score}_{\text{inter}}, \text{score}_{\text{info}}$

---

Jingqing Zhang, Yao Zhao, Mohammad Saleh, and Peter Liu. 2020. PEGASUS: Pre-training with extracted gap-sentences for abstractive summarization. In *International Conference on Machine Learning*, pages 11328–11339. PMLR.

Shiyue Zhang and Mohit Bansal. 2021. Finding a balanced degree of automation for summary evaluation. In *Proceedings of the 2021 Conference on Empirical Methods in Natural Language Processing*, pages 6617–6632, Online and Punta Cana, Dominican Republic. Association for Computational Linguistics.

Tianyi Zhang*, Varsha Kishore*, Felix Wu*, Kilian Q. Weinberger, and Yoav Artzi. 2020. BERTScore: Evaluating text generation with bert. In *International Conference on Learning Representations*.

Wei Zhao, Maxime Peyrard, Fei Liu, Yang Gao, Christian M. Meyer, and Steffen Eger. 2019. MoverScore: Text generation evaluating with contextualized embeddings and earth mover distance. In *Proceedings of the 2019 Conference on Empirical Methods in Natural Language Processing and the 9th International Joint Conference on Natural Language Processing (EMNLP-IJCNLP)*, pages 563–578, Hong Kong, China. Association for Computational Linguistics.

## A RECEVAL: Background and Details

In this section, we provide background for computing $\mathcal{V}$-information and describe additional implementation details of RECEVAL (Algorithm 1).

**Background on $\mathcal{V}$-Information** Let $X$ and $Y$ denote two random variables. Their conditional entropy is defined as $H(Y|X) = \mathbb{E}[-\log P(Y|X)]$ (Shannon, 1948). However, computing it requires knowledge of the true joint distribution of $X$ and $Y$ which can be infeasible in practice. As an alternative, Xu et al. (2020) propose $\mathcal{V}$-conditional entropy using a model family $\mathcal{V}$ that learns to map from $X$ to $Y$. It is defined as:

$$H_{\mathcal{V}}(Y|X) = \inf{}_{f \in \mathcal{V}} \mathbb{E}_{x,y \sim X,Y}(-\log f[x](y))$$

Each $f \in \mathcal{V}$ models the conditional distribution $P_f(Y|X)$. Thus, the model $\tilde{f} \in \mathcal{V}$, minimizing the above expectation, is optimized using a negative log-likelihood objective. Building on top of it, Xu et al. (2020) propose $\mathcal{V}$-*information* (also known as $\mathcal{V}$-*usable information*) which measures the amount of available information contained in $X$ about $Y$ that can be extracted using $\mathcal{V}$. It is defined as:

$$ I_{\mathcal{V}}(X \to Y) = H_{\mathcal{V}}(Y|\varnothing) - H_{\mathcal{V}}(Y|X) $$

Here, we denote the models used to compute $H_{\mathcal{V}}(Y|X)$ and $H_{\mathcal{V}}(Y|\varnothing)$ (minimizing expectation) as $g$ and $g'$ respectively.[4] Ethayarajh et al. (2022) propose *pointwise $\mathcal{V}$-information (*PVI*)* to measure the degree of usable information present in individual data points $(x, y)$ as:

$$ \text{PVI}(x \to y) = -\log g'[\varnothing](y) + \log g[x](y) $$

Similarly, conditional PVI relative to instance $z$ is defined as:

$$ \text{PVI}(x \to y|z) = -\log g'[z](y) + \log g[z, x](y) $$

At a high level, we use PVI to extract the amount of information present within and across reasoning steps, as discussed in detail in §4.1 and §4.3. Our use of PVI is consistent with Padmakumar and He (2021), who use a pointwise information metric to evaluate the relevance of summary sentences.

**Use of External Tools.** We use three categories of models: (i) Semantic Role Labeling (SRL) models for identifying RCUs, (ii) NLI models that measure entailment or contradiction in §4.1 and §4.2, and (iii) pretrained language models that form the model family $\mathcal{V}$ when computing PVI (in §4.1 and §4.3). To identify RCUs, we use out-of-the-box SRL models available in AllenNLP (Gardner et al., 2018; Shi and Lin, 2019) based on the BERT architecture (Devlin et al., 2019) (345M parameters). For detecting entailment or contradictions, we use a state-of-the-art NLI model (Laurer et al., 2022) with checkpoint available at Huggingface (Wolf et al., 2020).[5] We use the T5-large model (Raffel et al., 2020) as the model family $\mathcal{V}$ (770M parameters) finetuned on the gold reasoning chains

---

[4]Consistent with established notation in $\mathcal{V}$-information work, $f[x](y)$ denotes $P_f(y|x)$ where $f$ is a model. When $x = \varnothing$, we compute the probability of generating $y$ directly.

[5]NLI model available at: https://huggingface.co/MoritzLaurer/DeBERTa-v3-large-mnli-fever-anli-ling-wanli

(refer to paragraph below for details). Note that we use the original code for all text-generation metrics listed in §5.2. Specifically, rouge scores are computed using the python `rouge-score` package. To compute Somer's D correlation, we use the `somersd` function from the `scipy` package.

**RCU Computation.** As mentioned in §4.4, we use an SRL model to decompose a sentence into multiple 'subject-verb-object' frames. After obtaining a list of frames (often overlaping) from a sentence, we sort the frames by length and select a disjoint subset until any remaining frame is already contained in the sentence formed by the selected frames. From each frame, we remove modifiers (denoted by a separate tag) that contain a verb (checked using a PoS-tagging model from `nltk`) as it would also be identified as a separate frame. Once the RCUs are identified, we classify them into premise-RCUs or conclusion-RCUs based on the location in the sentence and rules based on the type of subordinating conjucntion (detected using PoS-tag). Typically, conclusion-RCU occurs at the very end of the sentence, but in case of 'because' or 'since' the RCU immediately following the conjunction is taken as the premise.

For instance, consider this example step from GSM-8K: "[ The boots cost \$5 more than both pairs of heels together ], so [ the boots cost 99 + 5 = \$104 ]." Here, the two RCUs are joined using "so" and thus the first RCU is the premise and the second is the conclusion. In a different example, "[ Allen's current age is 11/18*162 = 99 ] since [ the fraction of the ratio that represents Allen's age is 11/18 ]." Here, the first RCU is the conclusion and the second one is the premise based on the conjunction "since". Even if the sentence began with "since", we would identified the RCU immediately following it to be the premise.

**PVI Training.** Similar to Chen et al. (2022), we use the T5-large model (Raffel et al., 2020) as the predictive model family $\mathcal{V}$ that is finetuned on gold reasoning chains using the train split of each dataset (with dev splits used for model selection). However, in our case, the model is trained to generate the conclusion-RCUs or the entire reasoning step (instead of the label in a classification task as done in Ethayarajh et al. (2022); Chen et al. (2022)). We compute log-probability over the text sequence as the length-normalized average of log-probabilities over all tokens (Brown et al., 2020).

| Input Context ($\mathcal{X}$) | Gold Reasoning Chain | Orig. Perturbations | Our Perturbations |
|---|---|---|---|
| The moon is a kind of moon. Earth is a kind of planet. Moons orbit planets. Gravity causes orbits. What keeps the Moon orbiting Earth? | Moon orbits planets and earth is a kind of planet, so moon orbits earth. Gravity causes orbits, so gravity causes the moon to orbit the earth. **ROSCOE-SS Score:** 0.86 **RECEVAL Score:** 0.91 | Moon orbits planets and earth is not a planet, so moon orbits earth. Gravity causes orbits, so gravity causes the moon to orbit the earth. **ROSCOE-SS Score:** 0.24 **RECEVAL Score:** 0.21 | Moon orbits planets and earth is a kind of planet, so moon does not orbit earth. Gravity causes orbits, so gravity causes the moon to orbit the earth. **ROSCOE-SS Score:** 0.67 **RECEVAL Score:** 0.25 |
| Classifying means grouping objects by their properties. Shape is a property of appearance of an object. A galaxy is a kind of object. What feature is used to classify galaxies? | Classifying means grouping objects by their properties. Shape is a property of appearance of an object, so shape can be used to classify objects. A galaxy is a kind of object, so galaxies can be classified by shape. **ROSCOE-SS Score:** 0.89 **RECEVAL Score:** 0.84 | Classifying means grouping objects by their properties. Comets orbits are elliptical, so shape can be used to classify objects. A galaxy is a kind of object, so galaxies can be classified by shape. **ROSCOE-SS Score:** 0.31 **RECEVAL Score:** 0.18 | Classifying means grouping objects by their properties. Shape is a property of appearance of an object, so classification is a kind of process. A galaxy is a kind of object, so galaxies can be classified by shape. **ROSCOE-SS Score:** 0.58 **RECEVAL Score:** 0.22 |

Table 11: Differences in our perturbations to ones used in Golovneva et al. (2023) for errors NEG (top) and HALL (bottom). Overlapping text in input context and reasoning chains is underlined and perturbations are shown in red. For NEG with original perturbations, sentence embeddings of the perturbed overlapping sentence will be very different, leading to decrease in sentence similarity (does not occur in our perturbations). For HALL, shortcut is to check for facts missing from the input context by drop in sentence similarity (does not occur in our perturbations). This is also reflected in the ROSCOE and RECEVAL (intra-step) correctness scores for each reasoning chain.

For intra-correct$_{\text{PVI}}$, $g$ is a model trained to generate $y = \text{RCU}_c^{(i)}$ from $x = \text{RCU}_p^{(i)}$ and $g'$ is trained to generate $y = \text{RCU}_c^{(i)}$ directly. Using the train split of a reasoning dataset, we pool all steps from all reasoning chains. Each step is then decomposed into RCUs and constitutes one data point $(x, y)$ for training the aforementioned models. The input to the model (used to generate $y$) could be template, i.e. "[X] -> ", and "None -> ", or a natural language sentence, "[X], so ", and "So, " for $g$ and $g'$ respectively. Here, [X] represents the concatenated premise units $\text{RCU}_p^{(i)}$ (via 'and'). We find no significant change in performance when using the template or a natural language sentence. We use the latter to report performances in §6. For info-gain, the model $g$ is trained to generate $y = \hat{a}$ given $[z, x] = s^{(\leq i)}$ and the training data are partial reasoning chains conditioned to generate the predicted answer. Since input to $g'$ is $z = s^{(<i)}$, the input instances for $g$ and $g'$ overlap. Thus, we can use the same model for both $g$ and $g'$ as done by Chen et al. (2022). Note that $\hat{a}$ denotes the final answer sentence. So, $\hat{a}$ corresponds to the hypothesis sentence already provided in the EB dataset. In case of GSM-8K, we construct $\hat{a}$ by concatenating the question and the predicted answer, i.e., "[Q] Answer: [A]" where [Q], and [A] are placeholders for question and predicted answer respectively. Throughout training the hyperparameters used are:

learning-rate of $3e^{-5}$, 10 train epochs, with weight decay of 0.1 (all other hyperparameters are set to default). After training we select the model checkpoint (at epoch level) corresponding to the lowest 'rougeL' score on the dev split.

**Range of RECEVAL Metrics.** Our intra-correct$_{\text{entail}}$ and inter-correct scores fall in the range $[0, 1]$ where 0 indicates failure and 1 indicates perfect score. By construction, PVI can be positive, negative, or 0 which also applies to intra-correct$_{\text{PVI}}$ and info-gain$_{\text{PVI}}$. Positive PVI indicates a step is correct or informative, whereas negative (or zero) values indicate otherwise. Future works can explore normalization techniques to limit the range of these scores. Furthermore, informativeness of a step in a reasoning chain is an inherently subjective criterion that also depends on the underlying reasoning problem. Therefore, the info-gain$_{\text{PVI}}$ values of steps in different reasoning chains corresponding to different problem statements can be very different. Future work can also aim to address this variability.

**Downstream Performance on GSM-8K.** In §6.3, we use the FLAN T5-XXL model (11B parameters) to sample 20 diverse reasoning chains for each problem in the test set (with temperature of 0.7). Since both ROSCOE and RECEVAL contain multiple metrics, we use a simple aggregation strat-

You are given two types of phrases: a premise and a hypothesis, from a reasoning step. Based on the phrases, rate how well the premise entails the hypothesis on a scale of 0-1. 1 indicates perfect entailment and 0 indicates no entailment at all.

Premise: {premise-RCUs}
Hypothesis: {conclusion-RCU}
Score:

**Correctness Eval Prompt**

You are given a partial section of a reasoning chain and a model's predicted answer. On a scale of 0-1, rate how likely is the model to arrive at the answer based on the aforementioned steps. 0 indicates not at all likely and 1 indicates the answer directly follows from the steps.

Steps: {steps}
Answer: {predicted_answer}
Likelihood:

**Informativeness Eval Prompt**

Figure 3: Prompts used to compute correctness and informativeness metrics in RECEVAL with GPT-3.5-turbo.

egy for selecting reasoning chains. We select the chain with the highest scores on all metrics wherever possible. If such a chain does not exist, we rank chains based on each metric and select the chain with the lowest cumulative rank.

**Prompts used with GPT-3.5-tubro.** In §6.3, we described how to use RECEVAL with prompted LLMs. The prompts are shown in Figure 3 and were designed using a dev set of 10 reasoning chains from EB dataset.

## B Datasets and Errors

We expand on the dataset descriptions provided in §5.1, and explain various error types. A glossary of error types is present in Table 19.

### B.1 Entailment Bank

As described in §5.1, due to the construction of Entailment Bank, there is an overlap between $\mathcal{R}$ and $\mathcal{X}$. Therefore, if perturbations are applied to this overlapping information then it can spuriously lead to high correlation for any metric comparing $\mathcal{R}$ with $\mathcal{X}$ based on sentence-embeddings or $n$-grams. This happens because in gold or unperturbed chains there is high degree of overlap due to exact match and in the perturbed chains the overlap goes down significantly. However, if perturbations are applied to information not contained in $\mathcal{X}$, gold chains do not have high degree of overlap to begin with, and thus is a more challenging setting for evaluating metrics. Therefore, different from Golovneva et al. (2023), we only apply perturbations to facts/parts of the reasoning chain not in the input context.

We provide examples illustrating this phenomenon in Table 11. For negation errors, if we negate an overlapping source fact, comparing the chain with input the context leads to a direct drop

in sentence similarity. We remove this shortcut by negating facts not contained in the input context. For hallucination errors, if a source fact is hallucinated, one can detect hallucinations by simply checking if a source fact is missing (drop in cumulative sentence similarity when compared to $\mathcal{X}$). We remove this shortcut by only applying hallucination perturbations to intermediate facts not in $\mathcal{X}$. Additionally, instead of sampling hallucinated text from other reasoning problems, we sample hallucinated text from irrelevant sentences or distractors provided for each instance in Entailment Bank (Task 2). This leads to higher word overlap between hallucinated text and input context.

Perturbations are first applied to intermediate nodes in the reasoning tree and then converted into a natural language reasoning chain. While borrowing error types from Golovneva et al. (2023), we make the following three additional changes: Firstly, the hallucinated text is sampled from distractors. Secondly, swap errors are introduced between the intermediate node and its parents, so that we can ensure incoherence in the reasoning chain. Thirdly, repetition errors are implemented by repeating an intermediate node twice (parent of the second node is the first node). Instead of verbatim repetition, we also introduce adding a paraphrase using a Pegasus-based model (Zhang et al., 2020)[6] and an irrelevant but true sentence to the reasoning chain. So in case of Fig. 2(b), instead of verbatim repetition "the northern hemisphere is a kind of place", we would add text like "the norther hemisphere is a sort of location" and "daylight is when the sun shines" for PAR and RED errors respectively.

---

[6]Checkpoint: https://huggingface.co/tuner007/pegasus_paraphrase

| Input Context | Reasoning Chain |
|---|---|
| Tina buys 3 12-packs of soda for a party. Including Tina, 6 people are at the party. Half of the people at the party have 3 sodas each, 2 of the people have 4, and 1 person has 5. How many sodas are left over when the party is over? | [ Tina buys 3 12-packs of soda for 3*12=36 sodas ]. [ 6 people attend the party ], so [ half of them is 6/2= 3 people ]. [ Each of those people drinks 3 sodas ], so [ they drink 3*3=9 sodas ]. [ Two people drink 4 sodas ], which means [ they drink 2*4=8 sodas ]. With [ 1 person drinking 5 ], [ that brings the total drank to 5+9+8+3=25 sodas ]. As [ Tina started off with 36 sodas ], that means [ there are 36-25=11 sodas left ]. |
| Every day, Wendi feeds each of her chickens three cups of mixed chicken feed, containing seeds, mealworms and vegetables to help keep them healthy. She gives the chickens their feed in three separate meals. In the morning, she gives her flock of chickens 15 cups of feed. In the afternoon, she gives her chickens another 25 cups of feed. How many cups of feed does she need to give her chickens in the final meal of the day if the size of Wendi's flock is 20 chickens? | If [ each chicken eats 3 cups of feed per day ], then for 20 chickens [ they would need 3*20=60 cups of feed per day ]. If [ she feeds the flock 15 cups of feed in the morning ], and [ 25 cups in the afternoon ], then [ the final meal would require 60-15-25=20 cups of chicken feed ]. |

Table 12: Example of reasoning chains in GSM-8K dataset with identified RCUs shown as '[ . ]'.

| Input Context | Reasoning Chain | Error | ROSCOE-REP | RECEVAL |
|---|---|---|---|---|
| John has 3 boxes. Each box is 5 inches by 6 inches by 4 inches. The walls are 1 inch thick. What is the total inner volume of all 3 boxes? | Each box is 5*6*4 = «5*6*4=120»120 cubic inches. So they have a total of 120*3 = «120*3=360» 360 cubic inches. So the total volume is 360*1 = «360*1=360»360 cubic inches. Answer: 360 | REP | 0.46 | 0.12 |
| Mary is an avid gardener. Yesterday, she received 18 new potted plants from her favorite plant nursery. She already has 2 potted plants on each of the 40 window ledges of her large country home. Feeling generous, she has decided that she will give 1 potted plant from each ledge to friends and family tomorrow. How many potted plants will Mary remain with? | Mary has 2 potted plants on each of the 40 window ledges, for a total of 2*40 = «2*40=80»80 potted plants. She has 18 new potted plants, for a total of 80+18 = «80+18=98»98 potted plants. She is giving away 1 potted plant from each of the 40 window ledges, for a total of 1*40 = «1*40=40»40 plants. She will be remaining with 98-40 = «98-40=58»58 potted plants. Answer: 58 | None | 0.01 | 0.68 |

Table 13: Qualitative comparison between ROSCOE (REP) and RECEVAL (info-gain$_{PVI}$) scores on GSM-8K.

## B.2 GSM-8K

We directly use the human-annotated reasoning chains for GSM-8K collected by Golovneva et al. (2023). We refer readers interested in the data collection process, and details about each error type to Appendix F of their paper (c.f. Table 15). In Table 12, we provide some examples of gold (human-written) reasoning chains in GSM-8K along with our identified RCU annotations. Note that while EB-challenge is constructed such that a perturbed reasoning chain only contains one error at a time, errors in GSM-8K dataset can co-occur as it contains model-generated errors that can be diverse.

## C Additional RECEVAL Meta-Evaluation

**EB-Regular.** We evaluate the performance of all metrics on the originally perturbed sentences (EB-regular) in Table 16. While the relative trends between RECEVAL and other baselines remain the same, we find that ROSCOE's correlation values on HALL, NEG and SWAP are much higher than Table 1a where the aforementioned shortcuts do not exist. Furthermore, correlation values of text-generation metrics on HALL errors also decrease when spurious shortcuts are removed. Nevertheless, RECEVAL outperforms baselines on correctness errors. Note that we do not consider grammar, missing errors from Golovneva et al. (2023). This is mainly because missing steps involve a confounder and are hard to evaluate in a reference-free manner.

| Metric | HALL | NEG | SWAP |
|---|---|---|---|
| ROUGE-1 | 0.01 | 0.02 | 0.13 |
| ROUGE-2 | -0.01 | -0.02 | 0.14 |
| ROUGE-L | -0.04 | 0.01 | 0.10 |
| BERTScore | 0.09 | 0.02 | 0.07 |
| BARTScore | 0.00 | -0.01 | 0.07 |
| PRISM | 0.27$^\dagger$ | 0.03 | 0.08 |
| CTC Relevancy | 0.09 | -0.04 | -0.05 |
| CTC Consistency | 0.00 | -0.05 | -0.03 |
| ROSCOE-SA | 0.62$^\dagger$ | 0.40$^\dagger$ | 0.22$^\dagger$ |
| ROSCOE-SS | 0.34$^\dagger$ | 0.40$^\dagger$ | 0.09 |
| ROSCOE-LI | 0.20$^\dagger$ | 0.82$^\dagger$ | 0.16 |
| RECEVAL-intra-correct$_{entail}$ | 0.71$^\dagger$ | 0.86$^\dagger$ | 0.38$^\dagger$ |
| RECEVAL-intra-correct$_{PVI}$ | **0.89**$^\dagger$ | 0.14 | **0.39**$^\dagger$ |
| RECEVAL-inter-correct | 0.45$^\dagger$ | **0.88**$^\dagger$ | 0.22$^\dagger$ |

(a) Correctness

| Metric | REP | PAR | RED |
|---|---|---|---|
| ROUGE-1 | 0.45$^\dagger$ | 0.26$^\dagger$ | 0.15 |
| ROUGE-2 | 0.43$^\dagger$ | 0.21$^\dagger$ | 0.11 |
| ROUGE-L | 0.08 | 0.09 | 0.10 |
| BERTScore | 0.24$^\dagger$ | 0.16$^\dagger$ | 0.12 |
| BARTScore | 0.11 | 0.12 | 0.08 |
| PRISM | 0.15 | 0.11 | 0.09 |
| CTC Relevancy | 0.24$^\dagger$ | 0.14 | 0.10 |
| CTC Consistency | 0.25$^\dagger$ | 0.15 | 0.12 |
| ROSCOE-SA | **0.83**$^\dagger$ | _0.64_$^\dagger$ | 0.51$^\dagger$ |
| ROSCOE-SS | 0.81$^\dagger$ | 0.62$^\dagger$ | _0.54_$^\dagger$ |
| ROSCOE-REP | **0.83**$^\dagger$ | _0.64_$^\dagger$ | 0.48$^\dagger$ |
| RECEVAL-info-gain$_{PVI}$ | _0.66_$^\dagger$ | **0.68**$^\dagger$ | **0.67**$^\dagger$ |

(b) Informativeness

Table 14: Meta-evaluation (Somer's D) on EB-challenge (test). We **bold** the highest and underline the second-highest correlation (higher correlation is better). $^\dagger$Correlation values are statistically significant ($p < 0.05$).

| Metric | QUAL | COH | COM | FACT | HALL | RED | REP | LOGIC | MATH |
|---|---|---|---|---|---|---|---|---|---|
| ROUGE-1 | 0.12 | 0.20$^\dagger$ | 0.07 | 0.16 | 0.27 | 0.04 | 0.22 | 0.07 | 0.23 |
| ROUGE-2 | 0.09 | 0.14 | 0.06 | 0.10 | 0.17 | -0.02 | 0.56 | 0.03 | 0.11 |
| ROUGE-L | 0.17$^\dagger$ | 0.27$^\dagger$ | 0.19$^\dagger$ | 0.17 | 0.18 | 0.05 | 0.56 | 0.12 | 0.21 |
| BERTScore | 0.19$^\dagger$ | 0.23$^\dagger$ | 0.12 | 0.13 | 0.20 | 0.13 | 0.94 | 0.15 | 0.13 |
| BARTScore | 0.01 | 0.03 | -0.05 | 0.04 | -0.25 | -0.26 | 0.42 | 0.00 | -0.55$^\dagger$ |
| PRISM | -0.11 | -0.07 | -0.10 | -0.04 | -0.39 | -0.46$^\dagger$ | -0.09 | -0.17 | -0.34 |
| CTC Relevancy | -0.09 | -0.15$^\dagger$ | -0.08 | -0.11 | 0.01 | -0.37$^\dagger$ | 0.57 | -0.11 | -0.09 |
| CTC Consistency | -0.16$^\dagger$ | -0.20$^\dagger$ | -0.21$^\dagger$ | -0.13 | -0.01 | -0.32$^\dagger$ | 0.56 | -0.17 | -0.02 |
| ROSCOE-SA | 0.20$^\dagger$ | 0.19$^\dagger$ | 0.19$^\dagger$ | 0.08 | 0.22 | 0.39$^\dagger$ | 0.79 | 0.18$^\dagger$ | **0.44** |
| ROSCOE-SS | 0.20$^\dagger$ | 0.17$^\dagger$ | 0.17 | 0.14 | 0.25 | _0.51_$^\dagger$ | _0.87_ | 0.15$^\dagger$ | 0.23$^\dagger$ |
| ROSCOE-LI | 0.28$^\dagger$ | 0.26$^\dagger$ | 0.18 | _0.34_$^\dagger$ | 0.22 | 0.35 | **0.98** | 0.22$^\dagger$ | 0.09 |
| ROSCOE-REP | 0.20$^\dagger$ | 0.19$^\dagger$ | 0.19$^\dagger$ | 0.14 | 0.25 | _0.51_$^\dagger$ | _0.87_ | 0.18 | **0.44** |
| RECEVAL-intra-correct$_{entail}$ | **0.36**$^\dagger$ | 0.27$^\dagger$ | **0.21**$^\dagger$ | 0.24$^\dagger$ | _0.27_ | 0.21 | 0.63 | _0.23_$^\dagger$ | 0.18 |
| RECEVAL-intra-correct$_{PVI}$ | _0.34_$^\dagger$ | 0.27$^{\dagger\,\dagger}$ | _0.19_$^\dagger$ | 0.21$^\dagger$ | **0.28**$^\dagger$ | 0.10 | 0.46 | **0.25**$^\dagger$ | 0.24 |
| RECEVAL-inter-correct | 0.32$^\dagger$ | **0.31**$^\dagger$ | **0.21**$^\dagger$ | **0.37**$^\dagger$ | 0.26 | 0.40 | 0.63 | 0.22$^\dagger$ | 0.10 |
| RECEVAL-info-gain$_{PVI}$ | 0.30$^\dagger$ | _0.29_$^\dagger$ | _0.19_$^\dagger$ | 0.26$^\dagger$ | 0.26 | **0.55**$^\dagger$ | _0.87_ | 0.21$^\dagger$ | _0.32_ |

Table 15: Meta-evaluation (Somer's D) on GSM-8K (test). $^\dagger$Correlation values are statistically significant ($p < 0.05$).

Further, grammar issues that do not alter correctness can be measured easily by grammar-checking metrics used in ROSCOE-LC.

**Additional Baselines.** Tables 1 and 2 contain a subset of baselines used by Golovneva et al. (2023) as described in §5.2. We include additional text-generation baselines for EB and GSM-8K in Tables 14 and 15 respectively and explicitly indicate correlation values that are statistically significant. These include metrics such as ROUGE-1, ROUGE-L (Lin, 2004), PRISM (Thompson and Post, 2020), CTC Consistency (Deng et al., 2021). Furthermore, in Tables 14a and Tables 15 we also report performance of individual correctness metrics in RECEVAL, namely intra-correct$_{entail}$, intra-correct$_{PVI}$,

and inter-correct on the test splits. Note that in Tables 2 and 15, ROSCOE outperforms RECEVAL on REP errors. However, the relative frequency of REP errors is very low. Therefore, label imbalance results in spurious correlation between REP and overall coherency COH when using ROSCOE-LI.

# D  RECEVAL Correctness Metrics

In this section, we provide additional details and ablations about the correctness metrics in RECEVAL as discussed in §6.2.

**Oracle RCUs.** In §6.2, we evaluate our identified RCUs with gold RCUs using entailment trees from Entailment Bank. Given an intermediate node,

| Method | REP | HALL | NEG | SWAP |
|---|---|---|---|---|
| ROUGE-1 | 0.39 | 0.41 | 0.03 | 0.06 |
| ROUGE-2 | 0.36 | 0.39 | 0.11 | 0.09 |
| ROUGE-L | 0.21 | 0.19 | 0.01 | 0.23 |
| BERTScore | 0.26 | 0.41 | 0.15 | 0.17 |
| BARTScore | 0.03 | 0.06 | 0.08 | 0.18 |
| PRISM | 0.23 | 0.45 | 0.03 | 0.16 |
| CTC Relevancy | 0.26 | 0.06 | 0.03 | 0.04 |
| CTC Consistency | 0.31 | 0.16 | -0.05 | -0.02 |
| ROSCOE-SS (fine-tuned) | 0.51 | 0.51 | 0.54 | 0.04 |
| ROSCOE-SA (fine-tuned) | **0.82** | 0.85 | 0.92 | 0.61 |
| ROSCOE-LI | -0.04 | 0.40 | 0.91 | -0.05 |
| RECEVAL-correctness | 0.09 | **0.89** | **0.94** | **0.64** |
| RECEVAL-informativeness | 0.79 | 0.31 | 0.04 | 0.10 |

Table 16: Comparison of Somer's D correlation scores using baseline text-generation metrics, ROSCOE, and our metrics on perturbations to Entailment Bank by Golovneva et al. (2023).

| Method | HALL | NEG | SWAP |
|---|---|---|---|
| inter-correct | 0.14 | 0.90 | 0.16 |
| inter-correct (+ premises) | 0.15 | 0.87 | 0.13 |
| inter-correct$_{concat}$ | 0.14 | 0.89 | 0.22 |

Table 17: Comparison of different variants of inter-correct metric by including premises and concatenation instead of pair-wise comparison on dev split of EB-challenge.

| Method | $k$ | HALL | NEG | SWAP |
|---|---|---|---|---|
| inter-correct$_{no\text{-}contr.}$ | all | 0.14 | **0.89** | 0.22 |
| inter-correct$_{no\text{-}contr.}$ | 2 | 0.10 | 0.84 | 0.20 |
| inter-correct$_{entail}$ | 2 | 0.56 | 0.73 | 0.32 |
| inter-correct$_{PVI}$ | 2 | 0.84 | 0.10 | 0.34 |
| inter-correct$_{no\text{-}contr.}$ | 1 | 0.08 | 0.79 | 0.15 |
| inter-correct$_{entail}$ | 1 | 0.52 | 0.66 | 0.31 |
| inter-correct$_{PVI}$ | 1 | 0.81 | 0.05 | 0.26 |
| intra-correct$_{no\text{-}contr.}$ | 0 | 0.02 | 0.82 | 0.08 |
| intra-correct$_{entail}$ | 0 | 0.71 | 0.84 | 0.37 |
| intra-correct$_{PVI}$ | 0 | **0.86** | 0.16 | **0.38** |

Table 18: Comparison of different views of correctness based on current step and preceding $k$ steps on dev split of EB-challenge. Note that inter-correct$_{no\text{-}contr.}$ is same as inter-correct$_{concat}$.

step correctness when including premise-RCUs across all errors.

**Different views of correctness.** In §4.1 and §4.2, we present three views of correctness: (i) entailment, (ii) using PVI framework, and (iii) lack of contradictions. The first two are used to compute intra-correct and the last is used to compute inter-correct. As described in §4.1, correctness can be measured using various viewpoints (e.g., based on entailment or PVI). Then in Table 18 (bottom section), we compare all three views of correctness to compute intra-correct and conclude intra-correct$_{PVI}$, and intra-correct$_{entail}$ work best with hallucination and negate errors respectively (with comparable performance on swap). Thus, we conclude that intra-correct$_{entail}$ and intra-correct$_{PVI}$ have different degrees of effectiveness depending on the type of error and can be used in a complementary way. Now, we extend this analysis to evaluate how these three views of correctness compare when evaluating inter-step correctness. Since PVI and entailment variants concatenate information, to maintain uniformity, we use inter-correct$_{concat}$ for this analysis. We observe that the best performance on negation errors is obtained by inter-correct$_{no\text{-}contr.}$ with $k = all$, whereas for the rest best performance is obtained using intra-correct$_{PVI}$ ($k = 0$). Further, we find that inter-correct$_{PVI}$ works best to identify hallucinations (and swaps), whereas inter-correct$_{no\text{-}contr.}$ is best for negation across all values of $k$. Lastly, inter-correct$_{entail}$ correlates well across error types for different values of $k$. This leads to a unified correctness metric wherein different methods differ in the view of cor-

we decompose it into RCUs by picking the largest SRL frame (including modifiers). For the premise-RCUs, we find all RCUs from its parent nodes. This ensures that all the premise-RCUs used to form the conclusion are included when measuring correctness and avoids any irrelevant sentences (which are neutral when measuring entailment and independent from an information-theoretic perspective). This explains why using gold RCUs boosts the performance on intra-step-correctness.

**Variants of inter-correct.** As described in §4.2, we perform pair-wise comparison wit all prior information in $\mathcal{X}$ and conclusion-RCUs from preceding steps. Due to high overlap in information contained in premise-RCUs and $\mathcal{X}$, we did not measure correctness with respect to premises. Alternative to pair-wise comparison, one can also concatenate all prior information and check for contradiction directly (denoted by inter-correct$_{concat}$). We compare these three different implementations of inter-step correctness in Table 17. We find that the performance of concatenation and pair-wise variants is comparable across all error types. As expected, we observe similar performance of inter-

| Error | Dataset | Description | Correctness | Informativeness |
|-------|---------|-------------|:-----------:|:---------------:|
| HALL | All | Hallucinations: Step contains information not provided in the input context, could be irrelevant but makes the step wrong. | ✓ | ✗ |
| REP | All | For EB: Step contains verbatim repetition of information already in previous steps. For GSM-8K and DROP: Step contains verbatim repetition or paraphrasing of information already present. The step could be dropped without impacting correctness. | ✗ | ✓ |
| RED | All | Additional step in the reasoning chain containing information irrelevant to solving the problem. The information itself could be factual and consistent with input context. | ✗ | ✓ |
| PAR | EB | Additional step contains paraphrasing of information already in the reasoning chain. | ✗ | ✓ |
| NEG | EB | Compared to the gold chain, step contains negation of information altering the correctness. | ✓ | ✗ |
| SWAP | EB | Information within the step is swapped in order, altering the overall correctness. | ✓ | ✗ |
| QUAL | GSM-8K , DROP | Likert score (1-5), measures overall quality of reasoning chain and how well it answers the question. | ✓ | ✓ |
| COH | GSM-8K, DROP | Likert score (1-5), measures overall coherence of the reasoning chain, i.e. if it makes sense and is non-contradictory. | ✓ | ✓ |
| COM | GSM-8K, DROP | If the step contains any commonsense or general world knowledge related mistake. | ✓ | ✗ |
| FACT | GSM-8K, DROP | Step contains information that contradicts some information in the input context. | ✓ | ✗ |
| LOGIC | GSM-8K, DROP | Step contains errors in logical deduction, could be contradictory to previous steps or not enough support or evidence, relates to coherence. | ✓ | ✗ |
| MATH | GSM-8K | Arithmetic or math equation errors in the step. | ✓ | ✗ |

Table 19: Glossary of types of errors in EB-challenge and GSM-8K and how it relates to desired correctness and informativeness properties of good reasoning chains. Note that '✓' and '✗' denote the expected impact on correctness and informativeness in general. The actual impact depends on the reasoning chain and the exact error.

rectness employed and the number of preceding steps $k$ considered.

# E  Informativeness and Approximately Positive Information Gain (API)

Fig. 4 qualitatively shows how informativeness changes when adding a repeated (uninformative) step to gold reasoning chains in EB. As expected we see a sharp dip in our metric indicative of negative or minimally positive information gain.

**API.** In §6.2, we introduce API to quantify the trend of informativeness across steps in a reasoning chain. A reasoning chain is $\text{API}_k$ across steps if for every $k$ contiguous steps, these steps as a whole are more informative than the preceding steps. Based on the PVI framework, a reasoning chain would be $\text{API}_k$ if $\text{PVI}(s^{(i:i+k-1)} \to \hat{a}|s^{(<i)}) > 0, \forall s^{(i)} \in \mathcal{R}$. Below we show how to evaluate this quantity

directly in terms of our metric info-gain$_{\text{PVI}}$.

$$\text{PVI}(s^{(i:i+k-1)} \to \hat{a}|s^{(<i)})$$
$$= \log g[s^{(<i+k)}](\hat{a}) - \log g[s^{(<i)}](\hat{a}) \; (\because g = g')$$
$$= \log g[s^{(<i+k)}](\hat{a}) - \log g[s^{(<i+k-1)}](\hat{a})$$
$$\quad + \log g[s^{(<i+k-1)}](\hat{a}) \cdots - \log g[s^{(<i+1)}](\hat{a})$$
$$\quad + \log g[s^{(<i+1)}](\hat{a}) - \log g[s^{(<i)}](\hat{a})$$
$$= \text{PVI}(s^{(i+k-1)} \to \hat{a}|s^{(<i+k-2)}) + \cdots$$
$$\quad + \text{PVI}(s^{(i)} \to \hat{a}|s^{(<i)}) \; (\text{using definition in §4.3})$$
$$= \sum_{j=i}^{i+k-1} \text{info-gain}_{\text{PVI}}^{(j)}$$

**How does info-gain vary based on the number of preceding steps?** Finally, we are interested in analyzing the effect of the number of past steps conditioned on for computing info-gain. Instead of measuring the gain relative to all the preceding reasoning steps, we also consider using only $k$ preced-

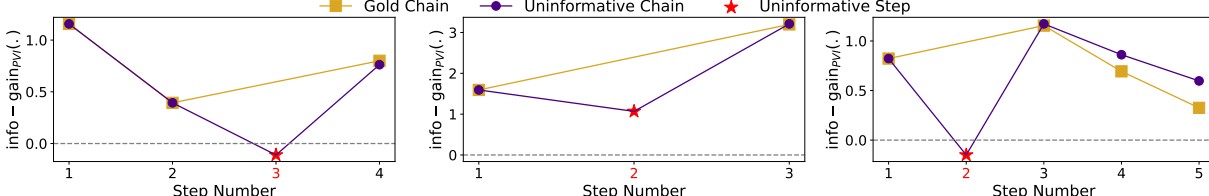

Figure 4: Trends in information gain of steps across gold and uninformative (REP) reasoning chains from EB-challenge. The position of the added uninformative step is highlighted in red on the x-axis and via '⋆' marker.

| Method | REP | PAR | RED |
|---|---|---|---|
| info-gain$_{\text{PVI}}$ ($k = 1$) | 0.65 | 0.66 | 0.64 |
| info-gain$_{\text{PVI}}$ ($k = 2$) | 0.70 | 0.69 | 0.68 |
| info-gain$_{\text{PVI}}$ ($k = all$) | 0.65 | 0.64 | 0.63 |

Table 20: Comparison of informativeness metric of RE-CEVAL on dev split of EB-challenge using different amounts prior steps ($k$) in the reasoning chain.

ing steps to compute information gain. In Table 20, we find that using $k = 2$ prior steps outperforms $k = 1$ consistently with nearly 0.04 higher correlation across error types. However, using all prior steps is comparable to $k = 1$ step. We suspect that the distinction between informative and uninformative chains becomes more pronounced when the reasoning chain is truncated and some of the required information for reasoning is absent from the context. Thus, we use $k = 2$ to compute info-gain in our final experiments in §6.1.