# OpenReview forum: "ReCEval: Evaluating Reasoning Chains via Correctness and Informativeness"
_EMNLP/2023/Conference — EMNLP 2023 Main_

### Official Review · Reviewer_qx1C · 2023-08-02

**Soundness:** 4

**Excitement:**

4: Strong: This paper deepens the understanding of some phenomenon or lowers the barriers to an existing research direction.

**Paper Topic And Main Contributions:**

This paper proposes a new reference-free metric suite, ReCEval, to evaluate the generated reasoning chains for QA tasks that require intermediate interpretable steps.

The authors consider two aspects that are crucial for the quality of reasoning chains, correctness and informativeness. The former measures how the generated chains can result in the correct answer while the latter aims at evaluating whether they contain useful and non-redundant information. Reasoning Content Units (RCUs) are used as the basic element in the metric which are "subject-verb-object" phrases extracted by a parser. Then the correctness is evaluated by the entailment relationship of RCUs within a single step or between a step and its preceding steps, as well as the stability given by an LM that a premise-RCU can lead to a conclusion-RCU. On the other hand, the informativeness is calculated by the LM probability of whether adding the current step can contribute to the gold answer.

Experiments are conducted on two datasets, Entailment Bank and GSM-8K, compared to n-gram baselines and embedding-based baselines, also the up-to-date metric ROSCOE. Results illustrate that ReCEval is generally better than other methods, while additional analyses are also involved to verify the effectiveness of each module as well as its variants.

**Questions For The Authors:**

A. Please check the Reason 1 in Reasons To Reject. How do you combine the three sub-metrics for the correctness of reasoning chains as the final correctness score? Or how do you select the proper one?

B. You mentioned you use T5 as the LM for PVI calculation, why do you select this model? Have you tried other models? Or T5 has the best performance among some candidates?

**Reasons To Accept:**

1. The proposed metric is basically well-motivated that can handle the demand of evaluating the quality of reasoning chains. To measure whether the obtained chains are accurate, ReCEval considers multiple dimensions including entailment and LM-based pointwise V-information within a step or between different steps.

2. I also admire that extensive experiments have been done in this paper, including ablations, settings using different hyper-parameters, and the potential of ReCEval to enhance the downstream reasoning chain generation. These experiments make the paper more comprehensive and credible.

**Reasons To Reject:**

1. I think maybe there are still some issues in the experiment part. 1) According to the original paper of ROSCOE, the authors use more than 10 datasets to prove the superiority of the proposed metric. But only two datasets are employed in this work, making it short of proof of generalization and expansibility. 2) Seems that the results in the main part about reasoning chain correctness only report the best performance among these three sub-metrics, i.e. intra-correct_{entail}, intra-correct_{PVI}, inter-correct. So how do you determine which one should be used in real applications? I did not notice any clear description. Or do you just select the best one according to its performance on a dev set? If so, the applicability of ReCEval will be largely lowered.

2. It is also a hot topic where LLMs like ChatGPT is utilized as a tool to make evaluation for NLP tasks. But there is no related content or discussion in this paper. I think it is easy work to call the APIs and check how they evaluate reasoning chains, which makes the paper or valuable.

**Reproducibility:**

4: Could mostly reproduce the results, but there may be some variation because of sample variance or minor variations in their interpretation of the protocol or method.

**Reviewer Confidence:**

3: Pretty sure, but there's a chance I missed something. Although I have a good feel for this area in general, I did not carefully check the paper's details, e.g., the math, experimental design, or novelty.

---

> ### Author Rebuttal · Authors · 2023-08-28
>
> Thank you for your review and acknowledging our comprehensive experiments and the downstream applications of ReCEval. Please refer to the following response to your comments.
>
> ### On choice of datasets:
> Entailment Bank and GSM-8k are two representative and diverse **multi-step** reasoning datasets. We did not choose other datasets because they were either
> * Similar:  MATH, ASDiv, and AQuA, are akin to GSM8K; however, only GSM8K dataset contains human evaluation of model-generated reasoning chains, whereas the rest only contain synthetic perturbations.
> * Synthetic: ProofWriter dataset involves reasoning over synthetically generated statements, on top of which there are synthetic perturbations to emulate errors.
> * Required only single-step rationales (like ESNLI and Cosmos).
>
> However, upon your suggestion, we compared the performance of Roscoe and ReCEval on the DROP dataset which also primarily consists of single-step rationales (<20% of samples needed multi-step rationales).
>
> | Method            | Qual | Coh | Com | Fact | Hall | Red | Rep | Logic |
> | :---------------- | :------: | :------: | :------: | :------: | :------: | :------: | :------: | :------: |
> | Roscoe-SA | 0.19 | -0.31 | *0.44* | *0.51* | -0.06 | -0.57 | -0.60 | 0.10 |
> | Roscoe-Rep  | 0.07 | **0.36** | -0.14 | 0.17 | 0.45 | *0.80* | **0.92** | 0.05 |
> | ReCEval-correctness  | **0.22** | *0.32* | **0.52** | **0.54** | **0.49** | 0.21 | -0.12 | **0.16** |
> | ReCEval-informativeness | *0.20* | **0.36** | 0.14 | *0.51* | *0.48* | **0.83** |  *0.89*| *0.12* |
>
> We find that ReCEval still exhibits superior performance compared to Roscoe. This goes to show that ReCEval can handle single-step rationales too. However, metrics designed specifically for inter-step correctness or informativeness across steps become less important in such scenarios.
>
>
> ### On sub-metric selection:
>
> This is an interesting point! When conducting meta-evaluation in Tables 1 and 2, we follow Golovneva et al. and report the highest correlation between submetrics to ensure fair comparison (refer to caption of Table 7 in Roscoe paper). We recommend using a combination of the correctness metrics in real applications such as getting downstream task performance improvement as discussed in Sec 6.3. Furthermore, in Appendix D, we provide additional discussion around the relationship between different correctness sub-metrics.
>
> In Sec 6.3, we employ a reranking strategy (described in Appendix A L1152-1162), allocating equal weights (0.25 each) to $intra\mbox{-}correct_{entail}$, $intra\mbox{-}correct_{PVI}$, $inter\mbox{-}correct$, and $info\mbox{-}gain_{LL}$. This straightforward approach proves effective in practice as shown in Table 8. In a scenario lacking gold reasoning chains, we can consider both correctness and informativeness to hold equal weight, assigning 0.25 weights to $intra\mbox{-}correct_{entail}$ and $inter\mbox{-}correct$, and 0.5 to $info\mbox{-}gain_{LL}$ (from Section 6.2, L533-L544). This approach, without using a dev set, enhances GSM-8k accuracy from 17.3% to 20.0%, maintaining a lead over Roscoe by at least 1 percentage point.
>
> As you rightly point out, having access to a dev set enables tuning these weights or even selecting the best correctness metric. In general, any method (including previous works such as Roscoe) that presents multiple metrics may encounter conflicting signals from various sub-metrics. Notably, ReCEval's categorization of metrics into correctness and informativeness  makes the metrics easier to interpret and enable simple yet effective combination strategies.
>
> ### On use of ChatGPT for calculating metrics:
> Thank you for your suggestion! Our contribution to the overall evaluation framework is complementary to the progress of LLMs. To show this, below we show results replacing our NLI/PVI modules with GPT-3.5-turbo on a subset of 50 reasoning chains from the EB dataset.
>
> **Experimental Setup:**
> To measure correctness, we prompt ChatGPT with the following instruction:
> ```
> You are given two types of phrases: a premise and a hypothesis, from a reasoning step. Based on the phrases, rate how well the premise(s) entail the hypothesis on a scale of 0-1. 1 indicates perfect entailment and 0 indicates no entailment at all.
> Premise: {premise-RCUs}
> Hypothesis: {conclusion-RCU}
> Score:
> ```
>
> To measure informativeness, we follow the steps in computing $info\mbox{-}gain_{LL}$ (Section 6.2, L533-L544). However, since the LLMs logits are not available, we prompt the model to output a real-valued score between 0 to 1 as the probability of generating the answer:
> ```
> You are given a partial section of a reasoning chain and a model's predicted answer. On a scale of 0-1, rate how likely is the model to arrive at the answer based on the aforementioned steps. 0 indicates not at all likely and 1 indicates the answer directly follows from the steps.
> Steps: {steps}
> Answer: {predicted_answer}
> Likelihood:
> ```
>
> These prompts were tuned on a dev set of 10 reasoning chains. Note that we use the above prompts for evaluating each individual step in a reasoning chain. Additionally, the correctness results use  $intra\mbox{-}step_{entail}$ as the pairwise comparisons used in inter-step correctness can be prohibitively expensive.
>
> **Results:**
> | Model            | Hall | Neg | Swap |
> | :---------------- | :------: | :------: | :------: |
> |SoTA finetuned NLI model | 0.89 | 0.88 | 0.39 |
> |GPT-3.5 turbo (zero-shot)| 0.86 | 0.91 | 0.38 |
>
> | Model            | Rep | Par | Red |
> | :---------------- | :------: | :------: | :------: |
> | $info\mbox{-}gain_{PVI}$ with T5-L | 0.65 | 0.66 | 0.64 |
> |GPT-3.5 turbo (zero-shot)| 0.54 | 0.58 | 0.56 |
>
> These results indicate that instead of using trained neural network modules (like finetuned NLI models or T5 models for PVI calculation), we can also prompt state-of-the-art LLMs to successfully measure correctness and informativeness. We'll include these results as well as a section discussing the concurrent research involving ChatGPT for model evaluation in our final version of our paper.
>
>
> ### On use of other model families for PVI:
>
> We select the T5 model in accordance with Chen et al. (2022), however in Sec 6.2 (L533-L544), we show ablation results using GPT-2 XL that yield comparable performance. Besides the ChatGPT results in the preceding paragraph, we further show that LLMs such as LLaMA-7B (without fine-tuning) can also be effectively used to measure informativeness. Please refer to our response to **reviewer HVRX**, paragraph **Different models for PVI computation when no gold reasoning chains are available**. In summary, we investigate three model families (T5, GPT-k, LLaMA), encompassing both open-source and gated LLMs, demonstrating the extensive spectrum of LMs that can be utilized and showcase the versatility of ReCEval.

---

### Official Review · Reviewer_HVRX · 2023-08-04

**Soundness:** 4

**Excitement:**

3: Ambivalent: It has merits (e.g., it reports state-of-the-art results, the idea is nice), but there are key weaknesses (e.g., it describes incremental work), and it can significantly benefit from another round of revision. However, I won't object to accepting it if my co-reviewers champion it.

**Paper Topic And Main Contributions:**

This paper proposes a reasoning chain evaluation metric called RECEVAL, which does not require ground truth. RECEVAL primarily assesses the correctness and informativeness of each step in the reasoning chain. The correctness is mainly evaluated based on the entailment relationship between premises and conclusions, while the informativeness is measured using the PVI (Pointwise Value of Information) information-theoretic method to determine whether the newly generated steps contribute new information towards the final answer. The paper conducts extensive experiments to validate the effectiveness of RECEVAL.

**Reasons To Accept:**

1. This paper has a clear structure and is well-written.
2. The experiments conducted in this paper are very extensive.
3. The evaluation metric issue addressed in this paper is of significant importance, and the viewpoints proposed by the authors are feasible.

**Reasons To Reject:**

My main concern is that the proposed evaluation metric in this paper is dynamic and relies on an external neural network, which may lead to the instability and irreproducibility of this metric. Specifically, I have the following concerns and questions:
1. Since neural networks may suffer from hallucination issues when generating text, how can we ensure that this problem does not arise when they are used as NLI models? In other words, how can we ensure that the neural network model can detect its own errors, which seems to be beyond its capability?
2. Is it feasible to concatenate all premises together for evaluating the relationship between conclusions and premises (intra-step correctness)? This manner seems different from traditional NLI tasks. What is the accuracy of this method, and have other approaches been attempted?
3. If a different set of training data is used for the model family V, how will it affect the evaluation? How should conflicting evaluation metrics be handled when two models trained on different datasets yield different results? What should be done if there is no gold reasoning chain available to train V for a particular task?
4. Have other models, such as LLM (Large Language Model) been attempted as the model family V?
5. There is a lack of case studies to validate the effectiveness of the proposed metric. It is particularly important to see comparative examples between existing metrics, the proposed metric, and human evaluations for reasoning types other than substitution on the Entailmentbank dataset, as well as examples from GSM-8K.


**Reproducibility:**

3: Could reproduce the results with some difficulty. The settings of parameters are underspecified or subjectively determined; the training/evaluation data are not widely available.

**Reviewer Confidence:**

4: Quite sure. I tried to check the important points carefully. It's unlikely, though conceivable, that I missed something that should affect my ratings.

---

> ### Author Rebuttal · Authors · 2023-08-28
>
> Thank you for your review and recognizing the importance of our proposed metric as well as acknowledging our extensive experiments. Please refer to the following responses to your insightful questions.
>
> ### Using neural network-based metrics:
>
> We acknowledge your point about using neural network models reliably as evaluators. We first want to note that the contribution of ReCEval goes beyond just the specifics of the models used for implementing the sub-metrics. ReCEval proposes a framework for evaluating reasoning chains based on two properties: correctness and informativeness. The paper shows various implementations for these and in the rebuttal, we also report results with LLMs such as LLaMA and GPT-3.5-turbo (paragraph **Different models for PVI computation …**).
>
> Next, on using model-based evaluators, we note that evaluating reasoning chains is a complex problem and even past work (including Roscoe and other baselines) have also relied on model-based metrics for solving this problem. Our specific implementation choices like the use of an NLI model are helpful in this regard, since these are supervised classification models and not generative models prone to hallucination. Furthermore, in ReCEval, we formulate evaluation as simpler tasks that models can solve fairly well, following established practices in interpretability literature. These steps are taken to mitigate concerns related to hallucination issues associated with model-based evaluators.
>
> ### Concatenating premises in NLI:
>
> We are not sure if we understood your question correctly but please let us know if this doesn’t help answer your question.
>
> As mentioned in Sec 4.1 (L254-256), concatenating all premise-RCUs within a step is feasible since steps typically contain just one or two premise-RCUs (refer to examples Fig 1 and 2, Table 10 and L1057-L1088 in Appendix A).
> The premise and conclusion RCUs are granular and typically, stand-alone units of information (e.g., `Earth is a kind of planet’) and hence we can concatenate the premise-RCUs to check if the conclusion-RCU is entailed by the combined piece of information obtained from concatenating the premise-RCUs. We believe that an NLI model is best suited for this task and good agreement scores also support this hypothesis.
>
> If you meant concatenation of all premise and conclusion units, when we try this concatenation across steps, we notice a decrease in correlation with errors for all error types (c.f. Table 1; EB test set).
>
> | Method            | Hall | Neg | Swap |
> | :---------------- | :------: | :------: | :------: |
> | ReCEval-correctness | 0.89 | 0.88 | 0.39 |
> | Concat. Premise, Conc Units | 0.30 | 0.72 | 0.10 |
>
> We believe this decline happens because when we concatenate across steps, unrelated premise-RCUs and conclusion-RCUs from other steps can act as distractors. This biases the NLI model toward the neutral label, even for gold reasoning chains. This approach might not be the intended or preferred usage of the NLI model, as it essentially asks the model to determine entailment for multiple pairs of premises and conclusions simultaneously.
>
> As alternative approaches, we also explored PVI models to compute correctness (as discussed in Sec 4.1, L262-274) or directly prompting an LLM like ChatGPT to generate a score (see **Incorporating Prompted LLMs within ReCEval** paragraph in response to **reviewer Yqzx**).
>
>
> ### Different models for PVI computation when no gold reasoning chains are available:
>
> We appreciate the comment that gold reasoning chains may not always be available. As mentioned in Section 6.2 (L533-L544), we acknowledge this challenge and explore an alternative approach using pretrained language models to compute log-probabilities, denoted as $info\mbox{-}gain_{LL}$. While this method is not an exact match to the PVI metric constraints by Ethayarajh et al. (refer to appendix A, L1000-1034), it still proves effective (c.f. Table 7). Below we also demonstrate that pretrained LLMs, including a stronger model like LLaMA-7B, can be employed within our ReCEval framework without requiring fine-tuning on individual datasets, yielding robust results.
>
> Method | Model | Rep | Par | Red
>  :--- | :--- | :---: | :---: | :---:
>  $info\mbox{-}gain_{PVI}$ | T5-L |  0.66 | 0.68 | 0.67
>  $info\mbox{-}gain_{PVI}$ | GPT-2 XL | 0.67 | 0.66 | 0.65
>  $info\mbox{-}gain_{LL}$ | GPT-2 XL | 0.58 | 0.60 | 0.60
>  $info\mbox{-}gain_{LL}$ | LLaMA-7B | **0.69** | **0.70** | **0.68**
>
> Please also refer to our response to **reviewer Yqzx** on how closed LLMs such as ChatGPT (GPT-3.5-turbo) can be incorporated into ReCEval with direct prompting.
>
>
> ### Qualitative Comparisons:
> Please refer to Appendix B for examples of datasets and errors. These details were mentioned in the appendix due to space constraints. However, as per your suggestion, we will add these to the main paper using the extra page in the final version. In the table below, we compare Roscoe and ReCEval scores using two examples shown in Table 9 from EB dataset.
>
> Method            | Problem | Gold Chain | Semantically Dissimilar Incorrect Chain | Semantically Similar Incorrect Chain | Error Type
>  :---------------- | :------: | :------: |  :------: | :------: | :------:
> Roscoe-ss | “moon orbits … the earth”  | 0.86 | 0.24 | 0.67 | Neg
> ReCEval (intra-correct) | “moon orbits … the earth”  | 0.91 | 0.21 | 0.25 | Neg
> Roscoe-ss | “classifying … galaxies?”  | 0.89 | 0.31 | 0.58 | Hal
> ReCEval (intra-correct) | “classifying … galaxies?”  | 0.84 | 0.18 | 0.22 | Hal
>
> Below is a similar table comparing Roscoe and ReCEval on a few selected examples from the GSM-8k dataset.
>
>  Problem | Reasoning Chain | Error Type | Roscoe (ss) | ReCEval (info-gain)
>  :------ | :------ |  :------: | :------: | :------:
> John has 3 boxes. Each box is 5 inches by 6 inches by 4 inches. The walls are 1 inch thick. What is the total inner volume of all 3 boxes? | Each box is 5\*6\*4=«5\*6\*4=120»120 cubic inches. So they have a total of 120\*3=«120\*3=360»360 cubic inches. So the total volume is 360\*1=«360\*1=360»360 cubic inches. Answer: 360 | Rep | 0.46 | 0.12
> Mary is an avid gardener. Yesterday, she received 18 new potted plants from her favorite plant nursery. She already has 2 potted plants on each of the 40 window ledges of her large country home. Feeling generous, she has decided that she will give 1 potted plant from each ledge to friends and family tomorrow. How many potted plants will Mary remain with? | Mary has 2 potted plants on each of the 40 window ledges, for a total of 2\*40 = «2\*40=80»80 potted plants. She has 18 new potted plants, for a total of 80+18 = «80+18=98»98 potted plants. She is giving away 1 potted plant from each of the 40 window ledges, for a total of 1\*40 = «1\*40=40»40 plants. She will be remaining with 98-40 = «98-40=58»58 potted plants. Answer: 58 | No error | 0.01 | 0.68
>
>
> Our observations from the shown examples reveal that Roscoe scores can be influenced by semantic similarity with the input context, i.e., incorrect steps are not semantically similar to the input. On the other hand, ReCEval metrics demonstrate the capability to detect errors even in intermediate steps that are semantically relevant to the problem. In the final version of the paper, we will incorporate qualitative examples for both EB and GSM-8K to illustrate these findings.

---

### Official Review · Reviewer_Yqzx · 2023-08-05

**Soundness:** 5

**Excitement:**

4: Strong: This paper deepens the understanding of some phenomenon or lowers the barriers to an existing research direction.

**Paper Topic And Main Contributions:**

This paper proposes ReCEval, a new metric that evaluates LLM reasoning chains based on their correctness and informativeness. Semantic Role Labeling is utilized to define reasoning chain units consisting of s-v-o triplets, and intra/inter-step correctness and informativeness are evaluated based on entailment or PVI (pointwise v-information). Compared with other reference-free metrics, ReCEval shows the highest correlation with error presence on Entailment bank and GSM8k

**Questions For The Authors:**

- How is PVI calculated using T5 model? These details are missing and should be included in the paper.
- Does this T5 model have to be finetuned for every dataset?


**Reasons To Accept:**

The proposed method is sound and makes effective use of existing approaches and resources in a novel manner.
The proposed method outperforms the previous state of the art (ROSCOE) in the space.
The paper is well written, and the ablation studies are informative.

**Reasons To Reject:**

While the paper makes it clear that existing reference-free metrics are inadequate to capture reasoning errors, this is not surprising. It would be more informative to include a large language model baseline such as [1], as these metrics would often be utilized for CoT reasoning with large language models (Section 6.3).

[1] Deductive Verification of Chain-of-Thought Reasoning. Ling et al., arXiv 2023

**Reproducibility:**

4: Could mostly reproduce the results, but there may be some variation because of sample variance or minor variations in their interpretation of the protocol or method.

**Reviewer Confidence:**

4: Quite sure. I tried to check the important points carefully. It's unlikely, though conceivable, that I missed something that should affect my ratings.

---

> ### Author Rebuttal · Authors · 2023-08-28
>
> Thank you for your review! We also appreciate you acknowledging the soundness and novelty of our method and the informative experiments! Please refer to the following responses to your questions.
>
> ### How is PVI calculated using the T5 model?
>
> Please refer to “**Background on V-Information**” and “**PVI training**” paragraphs in Appendix A (L1000-1034, and L1089-1134 respectively). As mentioned in L1025, PVI is defined as:
>
> $ PVI(x \rightarrow y) = - \log g’\[\phi\](y) +  \log g’\[x\](y) $
>
> i.e., PVI is the difference in log-likelihood of generating y given x and generating y directly.
> * **For calculating $\boldsymbol{intra\mbox{-}correct_{PVI}}$**, we train two T5-large models $g$ and $g’$. The first model ($g$) is trained to generate the conclusion-RCU when given the (concatenated) premise-RCUs of that step as input. The second model ($g’$) is trained to generate the conclusion-RCU directly with an empty string as input (L1093-1115).
> *  **For calculating $\boldsymbol{info\mbox{-}gain_{PVI}}$**, (conditional PVI, L1028), we only train one model ($g = g'$) to generate the predicted answer when given a partial reasoning chain as input (L1115-1128). The hyperparameters are mentioned in the appendix (L1129-1134).
>
> We use gold reasoning chains present in the train split of each dataset to train the model and use the dev set for model selection using the RougeL metric. We will bring back some of these details to the main paper using the extra page provided for the final version.
>
> ### Does this T5 model have to be finetuned for every dataset?
>
> This is an interesting question! To achieve the best results, we finetune the T5 model for both EB and GSM8K, since we have access to gold reasoning chains. However, since gold reasoning chains may not always be available, in Section 6.2 (L533-L544), we explore the use of pretrained LM to compute log-probabilities (described as $info\mbox{-}gain_{LL}$). Note that this is an approximation to the PVI metric, as it does not strictly follow the constraints listed by Ethayarajh et al. (refer to appendix A, L1000-1034). However, in Table 7, we find that while fine-tuned LMs are more effective, pretrained LMs can also be used to measure informativeness.
>
> In fact, as per the suggestion of the reviewers, we further experimented with an LLM (LLaMA-7B) to compute $info\mbox{-}gain_{LL}$ and observe that using a stronger language model surpasses a smaller LM finetuned on step distributions. See correlation results for different error types on EB dataset below (compare with Table 7). This shows that one can directly substitute an LLM into our ReCEval framework, without the need to fine-tune them on individual datasets and obtain strong results.
>
> Method | Model | Rep | Par | Red
>  :--- | :--- | :---: | :---: | :---:
>  $info\mbox{-}gain_{PVI}$ | T5-L |  0.66 | 0.68 | 0.67
>  $info\mbox{-}gain_{PVI}$ | GPT-2 XL | 0.67 | 0.66 | 0.65
>  $info\mbox{-}gain_{LL}$ | GPT-2 XL | 0.58 | 0.60 | 0.60
>  $info\mbox{-}gain_{LL}$ | LLaMA 7B | **0.69** | **0.70** | **0.68**
>
>
> ### Incorporating Prompted LLMs within ReCEval
>
> To show that we can incorporate prompted LLMs within our correctness and informativeness evaluation framework, we report results by prompting the GPT-3.5-turbo model on a subset of 50 reasoning chains from the EB dataset.
>
> **Experimental Setup:**
> To measure correctness, we prompt ChatGPT with the following instruction:
> ```
> You are given two types of phrases: a premise and a hypothesis, from a reasoning step. Based on the phrases, rate how well the premise(s) entail the hypothesis on a scale of 0-1. 1 indicates perfect entailment and 0 indicates no entailment at all.
> Premise: {premise-RCUs}
> Hypothesis: {conclusion-RCU}
> Score:
> ```
>
> To measure informativeness, we follow the steps in computing $info\mbox{-}gain_{LL}$ (Section 6.2, L533-L544). However, since the LLMs logits are not available, we prompt the model to output a real-valued score between 0 to 1 as the probability of generating the answer:
> ```
> You are given a partial section of a reasoning chain and a model's predicted answer. On a scale of 0-1, rate how likely is the model to arrive at the answer based on the aforementioned steps. 0 indicates not at all likely and 1 indicates the answer directly follows from the steps.
> Steps: {steps}
> Answer: {predicted_answer}
> Likelihood:
> ```
>
> These prompts were tuned on a dev set of 10 reasoning chains. Note that we use the above prompts for evaluating each individual step in a reasoning chain. Additionally, the correctness results use  $intra\mbox{-}step_{entail}$ as the pairwise comparisons used in inter-step correctness can be prohibitively expensive.
>
> **Results:**
> | Model            | Hall | Neg | Swap |
> | :---------------- | :------: | :------: | :------: |
> |SoTA finetuned NLI model | 0.89 | 0.88 | 0.39 |
> |GPT-3.5 turbo (zero-shot)| 0.86 | 0.91 | 0.38 |
>
> | Model            | Rep | Par | Red |
> | :---------------- | :------: | :------: | :------: |
> | $info\mbox{-}gain_{PVI}$ with T5-L | 0.65 | 0.66 | 0.64 |
> |GPT-3.5 turbo (zero-shot)| 0.54 | 0.58 | 0.56 |
>
> These results indicate that instead of using trained neural network modules (like finetuned NLI models or T5 models for PVI calculation), we can also prompt state-of-the-art LLMs to successfully measure correctness and informativeness. We'll include these results as well as a section discussing the concurrent research involving ChatGPT for model evaluation using the extra page in the final version of our paper.

---

### Meta-Review · Area_Chair_GRar · 2023-09-17

**Recommendation:** 5

**Metareview:**

The paper introduces "ReCEval," a novel metric designed to evaluate Large Language Model (LLM) reasoning chains based on their accuracy and informativeness. Using Semantic Role Labeling, reasoning chain units are defined as "subject-verb-object" triplets. The metric evaluates the correctness of these units through entailment relationships and uses the Pointwise Value of Information (PVI) to assess their informativeness. The paper emphasizes the importance of ensuring reasoning chains are both accurate and contain non-redundant, valuable information. Experiments were conducted on two datasets, Entailment Bank and GSM-8K. Compared to other metrics, including ROSCOE, ReCEval demonstrated superior performance, highlighting its potential in evaluating the quality of reasoning chains for QA tasks.

The paper presents a well-structured and novel approach to evaluating reasoning chains, outperforming previous state-of-the-art methods like ROSCOE. The method effectively leverages existing resources in a unique manner. The paper's clarity and comprehensive experimental design, including ablation studies, enhance its credibility. The evaluation metric addressed is of significant importance, and the solutions proposed by the authors are deemed feasible. The multi-dimensional approach of ReCEval, considering both entailment and LM-based pointwise V-information, is well-motivated and comprehensive. The extensive experiments, including various settings and hyperparameters, further validate the metric's potential to improve downstream reasoning chain generation.

Some weaknesses, discussed during the rebuttal, are listed below.
The paper's claim about the inadequacy of existing reference-free metrics is not surprising, and the inclusion of a large language model baseline is suggested for a more comprehensive evaluation. The dynamic nature of the proposed metric, relying on an external neural network, raises concerns about its stability and reproducibility. Questions arise about the neural network's ability to detect its own errors and the feasibility of concatenating all premises for evaluating intra-step correctness. The paper's experimental design is limited to only two datasets, questioning its generalizability. The method of selecting the best performance metric among the three sub-metrics is unclear, potentially limiting ReCEval's applicability.

---

### Decision · Program_Chairs · 2023-10-07

**Decision:**

Accept-Main

**Comment:**

The paper introduces "ReCEval," a novel metric designed to evaluate Large Language Model (LLM) reasoning chains based on their accuracy and informativeness. Using Semantic Role Labeling, reasoning chain units are defined as "subject-verb-object" triplets. The metric evaluates the correctness of these units through entailment relationships and uses the Pointwise Value of Information (PVI) to assess their informativeness. The paper emphasizes the importance of ensuring reasoning chains are both accurate and contain non-redundant, valuable information. Experiments were conducted on two datasets, Entailment Bank and GSM-8K. Compared to other metrics, including ROSCOE, ReCEval demonstrated superior performance, highlighting its potential in evaluating the quality of reasoning chains for QA tasks.

The paper presents a well-structured and novel approach to evaluating reasoning chains, outperforming previous state-of-the-art methods like ROSCOE. The method effectively leverages existing resources in a unique manner. The paper's clarity and comprehensive experimental design, including ablation studies, enhance its credibility. The evaluation metric addressed is of significant importance, and the solutions proposed by the authors are deemed feasible. The multi-dimensional approach of ReCEval, considering both entailment and LM-based pointwise V-information, is well-motivated and comprehensive. The extensive experiments, including various settings and hyperparameters, further validate the metric's potential to improve downstream reasoning chain generation.

Some weaknesses, discussed during the rebuttal, are listed below.
The paper's claim about the inadequacy of existing reference-free metrics is not surprising, and the inclusion of a large language model baseline is suggested for a more comprehensive evaluation. The dynamic nature of the proposed metric, relying on an external neural network, raises concerns about its stability and reproducibility. Questions arise about the neural network's ability to detect its own errors and the feasibility of concatenating all premises for evaluating intra-step correctness. The paper's experimental design is limited to only two datasets, questioning its generalizability. The method of selecting the best performance metric among the three sub-metrics is unclear, potentially limiting ReCEval's applicability.